# Digital measurement of SARS-CoV-2 transmission risk from 7 million contacts

Luca Ferretti[1,2,7 ✉], Chris Wymant[1,2,7], James Petrie[1,2], Daphne Tsallis[3], Michelle Kendall[4], Alice Ledda[5], Francesco Di Lauro[1,2], Adam Fowler[1,2], Andrea Di Francia[5], Jasmina Panovska-Griffiths[1,2,5], Lucie Abeler-Dörner[1,2], Marcos Charalambides[6], Mark Briers[6] & Christophe Fraser[1,2 ✉]

How likely is it to become infected by SARS-CoV-2 after being exposed? Almost everyone wondered about this question during the COVID-19 pandemic. Contact-tracing apps[1,2] recorded measurements of proximity[3] and duration between nearby smartphones. Contacts—individuals exposed to confirmed cases—were notified according to public health policies such as the 2 m, 15 min guideline[4,5], despite limited evidence supporting this threshold. Here we analysed 7 million contacts notified by the National Health Service COVID-19 app[6,7] in England and Wales to infer how app measurements translated to actual transmissions. Empirical metrics and statistical modelling showed a strong relation between app-computed risk scores and actual transmission probability. Longer exposures at greater distances had risk similar to that of shorter exposures at closer distances. The probability of transmission confirmed by a reported positive test increased initially linearly with duration of exposure (1.1% per hour) and continued increasing over several days. Whereas most exposures were short (median 0.7 h, interquartile range 0.4–1.6), transmissions typically resulted from exposures lasting between 1 h and several days (median 6 h, interquartile range 1.4–28). Households accounted for about 6% of contacts but 40% of transmissions. With sufficient preparation, privacy-preserving yet precise analyses of risk that would inform public health measures, based on digital contact tracing, could be performed within weeks of the emergence of a new pathogen.

Non-pharmaceutical measures such as social distancing, testing, contact tracing and quarantine are effective approaches to control the spread of epidemics, but they also entail significant social and economic costs[8,9]. It would be desirable to adjust these measures throughout an epidemic as epidemiological understanding increases or as the pathogen evolves. Optimization of such interventions requires methods to quantify transmission risk.

Despite the large amount of SARS-CoV-2 data collected globally, quantitative risk assessments at the level of individual exposures have been limited to a few large-scale, manual contact-tracing studies[10,11]. Another approach is provided by contact-tracing apps on smartphones, which were implemented for COVID-19 in many countries. These apps digitized the process of contact tracing based on recording close-proximity events between smartphones[1] and performing quantitative risk assessment by measuring proximity[3,12,13] and duration of exposure to cases, although their real-life accuracy has been questioned[14–17]. Contact-tracing apps are useful for public health if they are able to estimate the risk of pathogen transmission, and should be evaluated to improve their functionality and ensure public trust[2,18].

For contact tracing and, more generally, for distancing guidelines, public health authorities worldwide often used a binary classification of risk—for example, whether individuals spent 15 min or more at a distance of 2 m or less from a case[4,5]. Contact-tracing apps were calibrated to approximately match these heuristic rules. In the UK, which experienced a large-scale epidemic and implemented a substantial test-and-trace infrastructure, this advice led to more than 20 million notifications and quarantine requests from manual[19] and digital[20] contact tracing, with a peak of over 1.5 million per week in July 2021. The socioeconomic costs could have been significantly mitigated by improved, fine-tuned guidelines for contact tracing and quarantine. Doing this would require two components: (1) data and methods for quantitative assessment of how the probability of transmission varies with different factors, and (2) tools to measure those risk factors for contacts, to estimate their individual level of risk and respond appropriately.

Digital contact tracing in England and Wales was implemented through the National Health Service (NHS) COVID-19 app[6], which was active on 13–18 million smartphones each day during 2021 (ref. 7). The app recorded measurements of the proximity and duration of exposure to an index case using the privacy-preserving Exposure Notification framework[21], with custom analysis of Bluetooth signal attenuation between smartphones to estimate proximity[22]. By relating these data to

[1]Pandemic Sciences Institute, Nuffield Department for Medicine, University of Oxford, Oxford, UK. [2]Big Data Institute, Li Ka Shing Centre for Health Information and Discovery, Nuffield Department for Medicine, University of Oxford, Oxford, UK. [3]Zühlke Engineering Ltd, London, UK. [4]Department of Statistics, University of Warwick, Coventry, UK. [5]UK Health Security Agency, London, UK. [6]The Alan Turing Institute, London, UK. [7]These authors contributed equally: Luca Ferretti, Chris Wymant. ✉e-mail: luca.ferretti@bdi.ox.ac.uk; christophe.fraser@bdi.ox.ac.uk

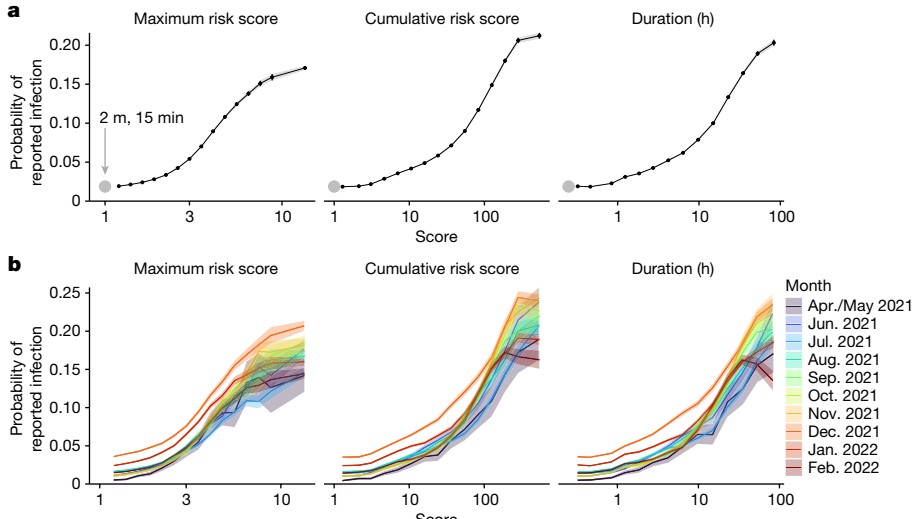

**Fig. 1 | App risk score and duration of exposure correlate with probability of infection. a**, The probability of reported infection—that is, the probability of a contact reporting a positive test through the app shortly after receiving an exposure notification—as a function of three summary metrics of their exposure measurements (scores): (1) maximum risk score from any exposure window (each lasting 30 min), (2) cumulative risk score, summed over all exposure windows and (3) total duration of the exposure, summed over all exposure windows. Grey points denote our estimates for the probability of reported infection after 15 min at 2 m distance from an individual with standard infectiousness. Black points indicate the bins used for the risk predictor. **b**, Probability of reported infection disaggregated by month of notification. Central values correspond to maximum-likelihood estimates, shading and (small) whiskers indicate 95% confidence intervals ($n = 7,047,541$ contacts). Tabulated values can be found in Supplementary Tables 6 and 7.

whether the exposed individual subsequently reported a positive test through the app, we showed how the probability of SARS-CoV-2 transmission varied with app-recorded measurements. We analysed 7 million exposure notifications from April 2021 to February 2022 comprising 23 million hours of cumulative exposure and 240,000 positive tests reported following notification. We demonstrate that the NHS COVID-19 app accurately translated proximity and duration of exposure into a meaningful epidemiological risk score, and we quantify how these factors affected the actual probability of transmission.

We use the terms 'case' to mean an individual whose infection was confirmed by testing, 'index case' to mean a case who triggered a contact-tracing process and 'contact' to mean an individual identified as having had some level of exposure to an index case (including, in general, individuals whose level of exposure is evaluated as being below a particular risk threshold).

The NHS COVID-19 app assessed the transmission risk for a contact by partitioning the full exposure event into a set of non-overlapping 'exposure windows', each lasting at most 30 min. For each window the app calculated a risk score[23,24]:

$$\text{Risk score} = \text{proximity score} \times \text{duration within } 30\text{-min window} \times \text{infectiousness score}$$

The proximity score was constant below 1 m and decreased as the inverse square of the distance if greater than 1 m. A scaling of risk in proportion to duration follows from microbial risk assessment expectations. Infectiousness was scored as either 'standard', 'high' (2.5×) or zero depending on the timing of exposure relative to the index case symptom onset date (or positive test date when no symptom onset was recorded)[23,25]. For ease of interpretation we normalized risk score such that it equals 1 for an exposure at 2 m distance from an index case with standard infectiousness for 15 min (that is, the typical threshold for manual contact tracing), implying a maximum possible score of 20.

Contacts were notified of a risky exposure if they had at least one exposure window with a risk score exceeding the threshold for notification, which was 1.11 with our normalization (Extended Data Fig. 1 shows the threshold in distance–duration space). When a contact was

notified, their app sent anonymous exposure data to the central server. These data were then sent in separate, unlinked data 'packets', one for each exposure window that had a risk score over the notification threshold (about half of the contacts had more than one exposure window; Extended Data Table 1). These packets formed the basis for our analysis: we analysed only contacts who were notified and had at least one exposure above the risk threshold. We grouped windows that were likely to have come from the same contact as a recording of the whole exposure history between that contact and the associated index case (excluding windows below the notification threshold). If a given individual was notified multiple times during our study, each notification was treated as though it were of a separate contact owing to the absence of unique identifiers.

The data also indicated whether the contact had reported a positive SARS-CoV-2 test through the app during an interval beginning with their notification and ending 14 days after exposure. The fraction of contacts doing so defines the probability of reported infection. This is a proxy for the true probability of being infected, although it is significantly underestimated: an unknown but probably appreciable fraction of infected app users either did not seek a test, or did not report their positive result through the app, or reported it outside of the aforementioned interval. As a reference, the number of infections in adults in the same period in the UK was two to three times greater than the number of cases[26].

The linkage between exposure measurements and reported test positivity enables apps to be used for precision epidemiological estimation while preserving privacy. We analysed how contacts' exposure data, recorded in separate 30-min windows, can predict their probability of reporting a positive test following their exposure. The peak risk experienced by an individual can be summarized by the maximum risk score measured by the app among all of their 30-min exposure windows. This summary metric is what the app actually used: contacts were notified only when it was above the threshold. We found an increasing probability of reported infection as maximum risk score increased (Fig. 1a). This pattern holds irrespective of season or epidemic wave (Fig. 1b). This simple analysis demonstrates that the approach used by the app to calculate risk correlates with the actual risk of transmission.

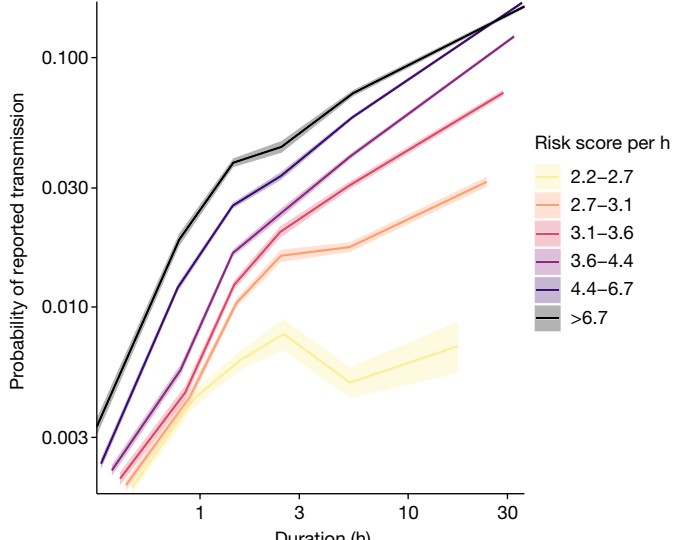

**Fig. 2 | The probability of transmission is affected by both duration and proximity as captured by risk score.** log–log plot of the probability of reported transmission—that is, the probability that the contact reported a positive test that we attributed to the transmission event traced, as a function of the binned duration of exposure and mean risk score per hour (that is, cumulative risk score divided by duration). Solid lines connect maximum-likelihood estimates for each bin and shading around these denotes 95% confidence intervals. Tabulated values can be found in Supplementary Table 8.

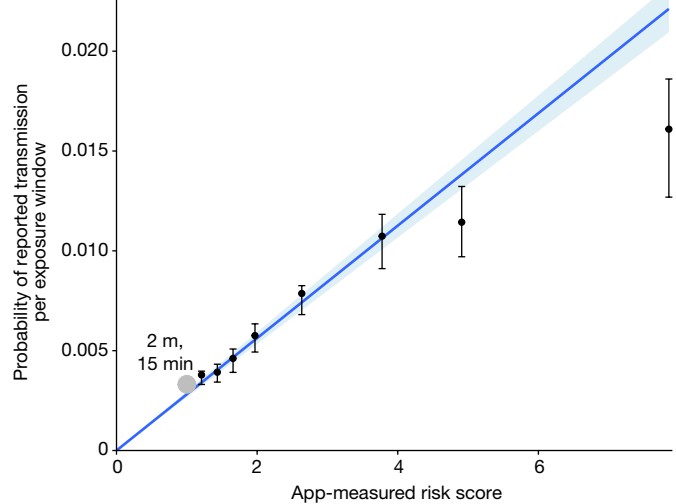

**Fig. 3 | Transmission probability per exposure window increases almost linearly with risk score.** The probability of reported transmission per exposure window—that is, the estimated probability of transmission in an individual 30-min exposure window followed by reporting of a positive test—as a function of the app-measured risk score for that window. Points show the maximum-likelihood estimate ($n$ = 2,507,879 contacts) and error bars on points indicate 95% confidence intervals. We fit a weighted robust linear regression without intercept to the points, with shading around the line indicating 95% confidence intervals in its gradient, highlighting that the probability of reported transmission is proportional to the app-measured risk score. The grey point denotes our estimate for the probability of reported transmission after 15 min at 2 m distance from an individual with standard infectiousness. Tabulated values can be found in Supplementary Table 9.

We defined two further summary metrics of risk measurements for each contact: total duration of exposure and cumulative risk score, both aggregated over all exposure windows from the contact. Both of these metrics are more discriminatory than the maximum risk score. The probability of reported infection continues to increase as the duration and cumulative risk increase, even after several days of cumulative exposure (Fig. 1).

These results suggest that both the instantaneous level of risk and duration of exposure affect the risk of transmission. We also expect a background level of risk from exposures either not recorded or not reported by the app; we estimated this level by statistically modelling it as proportional to the local risk of infection among app users at that time (Methods). We therefore stratified contacts by two summary metrics of their app-recorded measurements simultaneously: the duration of their exposure and their mean risk score per unit time. For each stratum of contacts we calculated the fraction reporting a positive test through the app during the observation window, as previously, now also subtracting the estimated background risk; we refer to the resulting quantity as the probability of reported transmission. (This differs from the probability of reported infection in that the background has been subtracted, and thus we attribute transmission to the exposures measured by the app. Both of these probabilities are lower than the corresponding true probabilities owing to incomplete reporting.) As expected, we found that the level of risk measured by the app and the duration of exposure both contribute to the probability of reported transmission (Fig. 2). Duration is the more important predictor. For short exposures the probability of reported transmission grows linearly with duration at a rate of 1.1% per hour, increasing sublinearly only after a few hours (Extended Data Fig. 2).

These results suggest that overall risk is determined by contributions from each separate exposure window, with greater contributions from riskier windows, in addition to background risk. To disentangle these effects we used a statistical model for combined contributions to overall risk, estimating the separate contributions from each window and from the background. We refer to these separate contributions from each exposure window as the probability of reported transmission per exposure window. We found that the probability of reported transmission per exposure window was proportional to the app's risk score for that window with remarkable accuracy, increasing by 0.3% per unit, providing validation that the app's risk calculation is epidemiologically meaningful. Figure 3 shows this relationship for exposures lasting between 1 and 3 h. The relationship is robust with respect to individual heterogeneities or under-reporting of positive tests among contacts (Extended Data Fig. 3).

Heterogeneities in the context of an exposure are expected to have a major effect on transmission risk. The context is not recorded by the app, but date and geographical area may be correlated with context and other causal factors. As an example, the probability of transmission from low-risk exposures is higher over the weekend than on weekdays (Extended Data Fig. 4), and it appears to be lower in London and other conurbations than in rural and urban areas (towns and cities), particularly at the lower end of the risk spectrum (Extended Data Fig. 4).

The impact of transmission control measures that target risk factors is determined by the distribution of these factors in the population, as well as how predictive they are of risk. Figure 4a–c shows population distributions over contacts of maximum and cumulative risk scores and total duration of exposure. We show the distributions separately for (1) all contacts and (2) transmissions—that is, only those contacts who reported a positive test result through the app in the observation window, for whom we attributed the infection to the recorded exposure. Distributions are strongly left-skewed, with low risk scores and short durations most common among contacts, in agreement with previous observations in specific contexts such as university campuses[27]. Larger risk scores and longer durations are seen disproportionately more for transmissions than for all contacts, in keeping with our earlier results and mechanistic understanding of pathogen transmission risk.

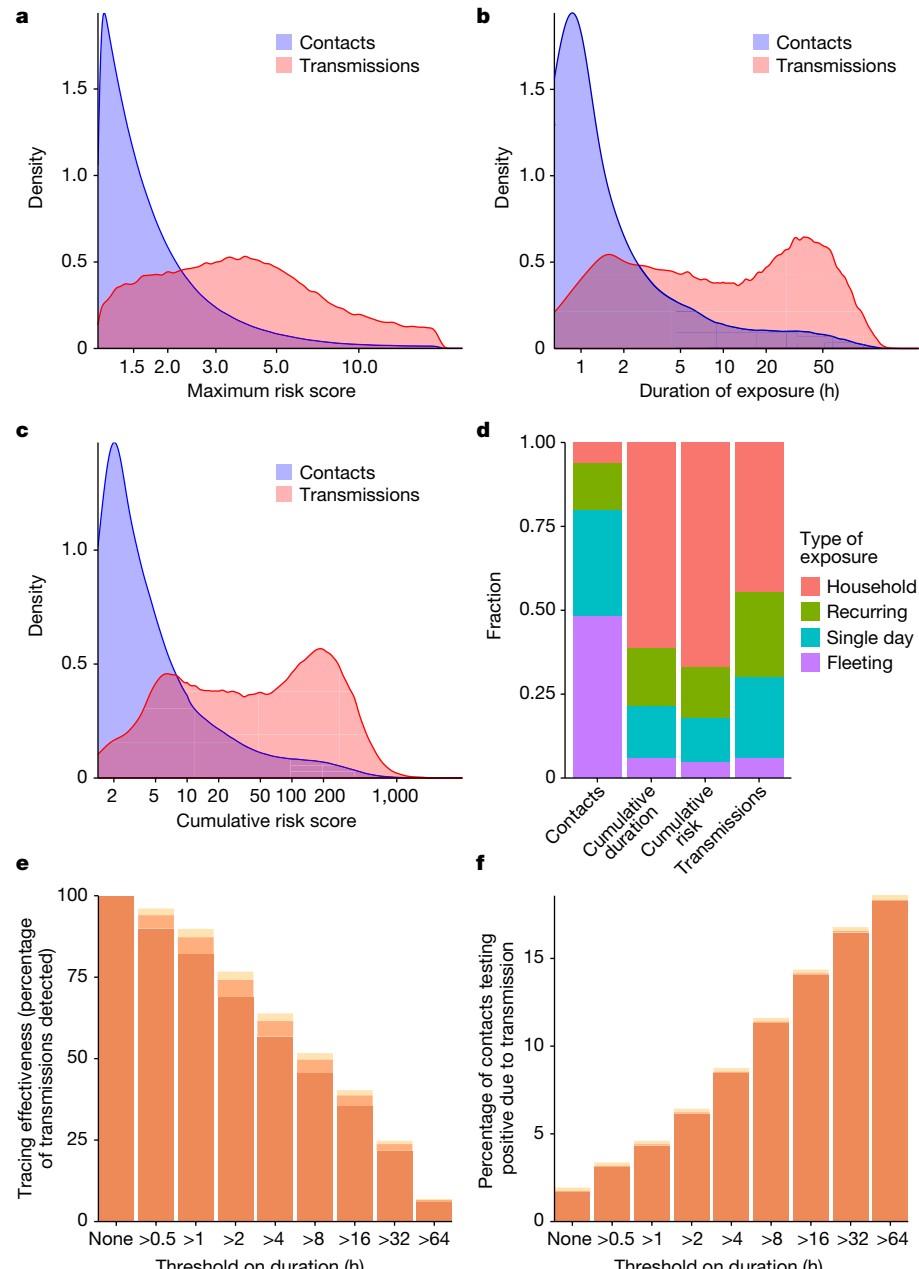

**Fig. 4 | Short, intermediate and long exposures all contribute to SARS-CoV-2 transmissions in the population. a**–**c**, Distributions over contacts of summary metrics for their app-recorded exposure measurements, shown separately for all contacts in the dataset (all of whom were notified, shown in blue) and for 'transmissions'–that is, only those contacts who reported a positive test result through the app in the observation window, for whom we attributed the transmission to the recorded exposure rather than the background risk (shown in red). **a**, Distribution of maximum risk score. **b**, Distribution of duration of exposure. **c**, Distribution of cumulative risk score over all exposure windows. **d**, Categories of contacts reflecting the context of their exposure. The first (far left) bar shows the fraction of contacts in each category; the other bars show the fractions of overall cumulative duration of exposure, cumulative risk score and number of transmissions associated with each category. **e**, Fraction of all traced transmissions that would still have been traced if only contacts with exposures longer than a given duration had been traced. This relative effectiveness of contact tracing at different thresholds corresponds also to the reduction in the reproduction number (Rt) in a counterfactual scenario with a higher notification threshold relative to reduction in Rt in the factual scenario. **f**, Fraction of contacts becoming infected during the recorded exposure and reporting a positive test–that is, the ratio of transmissions to contacts–among all contacts with exposures longer than a given duration. **e**,**f**, Shading at the top of the bars denotes 95% confidence intervals from uncertainty on background risk. Tabulated values can be found in Supplementary Table 10.

Across all contacts, most exposures are brief (median duration 40 min) yet most detected exposures that result in transmission last several hours (median duration 6 h; 82% last more than 1 h; Fig. 4e), suggesting that contact tracing for SARS-CoV-2 would retain more than 80% of its effectiveness if applied with a threshold of 1 h. Cumulative risk and duration show a bimodal distribution for transmissions; duration has a wide distribution (interquartile range 1.4–28 h), with a peak at around 1–2 h of exposure and another at around one to two full days of cumulative exposure, the latter most probably corresponding to household contacts.

To clarify the contribution of different exposure patterns and contexts to SARS-CoV-2 spread, we classified contacts into four categories intended as an approximate reflection of different contexts: contacts exposed for at least 8 h in one day (household contacts), non-household

contacts with recurring exposures on multiple days, contacts exposed during a single day (between 30 min and 8 h) and fleeting contacts (less than 30 min). Household and recurring contacts accounted for 6 and 14% of all app-recorded contacts but were responsible for 41 and 24% of transmissions, respectively (Fig. 4d). The long duration of household exposures—33 h on average—and their closer proximity explain their disproportionate role in transmissions (Extended Data Table 2).

How effective are these app-measured predictors in binary risk classification for contacts? Figure 4e,f shows the sensitivity–specificity trade-off among contacts arising from the use of different thresholds on duration. Extended Data Fig. 5 shows the trade-off for several predictors, including machine learning classifiers using binned counts of risk scores and extra information such as background risk, date and region. There was a small improvement in classification by using duration or cumulative risk rather than maximum risk, and the only significant further gain came from the inclusion of background risk. In fact, duration and background risk alone were sufficient for a near-optimal prediction, with an area under the receiver operating characteristic curve of 0.73.

These quantitative risk measurements enable optimization of a variety of management strategies based on simple and effective predictors such as duration of exposure to a case. As an example, we previously proposed milder 'amber' notifications as an alternative to quarantine for intermediate-risk contacts during the pandemic[1,28], and these were implemented in some settings[29]. If amber notifications were optimally assigned for intermediate durations of exposure, pursuing an optimized strategy of PCR testing following an amber notification could reduce the socioeconomic costs of an illustrative intervention by 30–50% for a similar epidemiological impact (Extended Data Fig. 6), or increase its effectiveness by 30–50% for similar costs (Extended Data Fig. 7).

## Discussion

We performed a large-scale study of how SARS-CoV-2 transmission probability varies with app-recorded risk measurements of the proximity and duration of exposures, analysing data from 7 million contacts notified by the NHS COVID-19 app in England and Wales. We found that the probability of infection strongly correlated with duration of exposure, as well as with the maximum and cumulative risk scores measured by the app. As a measure of proximity, the app's risk score for individual exposure windows captured the relative probability of transmission with remarkable accuracy. Furthermore, the app-measured cumulative risk score was the best single predictor of probability of transmission among those tested, in agreement with expectations from microbial risk modelling (Supplementary Methods section 1.5). This provides highly encouraging validation for the risk modelling underlying the NHS COVID-19 app[23,30] and for future development of similar tools.

Our results have immediate implications for contact tracing. We found that the cumulative duration of exposure to infected individuals was a key predictor of transmission in the COVID-19 pandemic, and needs to be accounted for in preparation for future epidemics of respiratory pathogens. Because duration of exposure to known cases can usually be recalled without the support of digital tools, it could be immediately incorporated into manual contact-tracing interviews. Contacts should be notified and managed based on duration of exposure as well as other risk factors; knowledge transfer should prove relatively easy—for example, through automated tools to support manual contact-tracing staff with their interview-based risk assessment. Beyond identification of predictors of infection, our quantitative risk measurements also enable optimization of different public health outcomes and epidemic management strategies such as amber notifications and postexposure prophylaxis.

One result of particular importance beyond contact tracing is our empirical demonstration of the continuing increase in probability of transmission with the duration of exposure to an infected individual. Spending a long time at greater distance from an infected person carries similar risk to shorter times at smaller distances. 'Physical distancing' strategies to reduce risk should therefore consider the relevance of both time and space. The continued increase in risk that we observed over multiple days shows that individuals can still benefit by beginning precautionary measures even after having already spent days exposed to an index case—for example, in the same household.

The effectiveness of epidemic control measures depends on the population distribution of risk. Exposures are highly skewed towards short and low-risk encounters; on the other hand, transmissions are caused by exposures in a wide range of risk, with duration varying from 1 h to several days. Our results can pave the way towards more targeted and graded interventions that account for the varying frequency and risk of different exposures.

The main limitation of our analysis is the absence of data on the context of an exposure: setting, immunity, level of ventilation and so on. The observed risks we report are averages over these unknown factors. Some of these factors might affect the risk score recorded by the app and the true risk in different ways—for example, being indoors is linked to poorer ventilation, which increases true risk but not risk score. Manual tracing can obtain contextual data through interviews; in practice these data are sometimes used to assess risk, but they should be collected more systematically to build a more informed classification of risk. Recording of direct or indirect information on the context of exposures, either through the app (for example, by implementing indoor/outdoor detection) or linking it from external sources, could significantly improve risk assessment.

Another limitation of our study is the inclusion of exposures only when their risk score crossed the app's notification threshold, excluding transmissions resulting from a large number of very-low-risk exposures. These transmissions are likely to have a role in the spreading of SARS-CoV-2 in specific settings but are unlikely to be a major driver of the epidemic. Also, because testing was not compulsory for contacts, infections were probably under-reported and absolute transmission rates must be interpreted with caution. Biases in testing or reporting, such as increased propensity to get tested after learning that a close contact tested positive, could also have affected our results.

In summary, if deployed at scale, contact-tracing apps for infectious diseases have potential not only as interventions to reduce transmission[6,7] but also as tools to develop quantitative epidemiological understanding. Doing this and translating it into improved interventions takes time. We should strive to accelerate and improve this process as a key step toward preparedness for future epidemics. Tools and methods for quantitative risk measurement and assessment should be further developed and integrated into the public health toolbox for the benefits they can bring now and in readiness for rapid deployment at the start of the next pandemic.

Recent decades have seen increasing focus on 'personalized' or 'precision medicine': using an individual's biomarkers to inform their treatment and disease prevention. Epidemiological interventions that are concerned with population health, based on exposures and risks, have a long way to go to catch up. Nevertheless, the benefit of doing so is clear: dynamically tailoring responses according to individual risks measured at scale could turn blunt instruments into sharp ones. Digital contact tracing and the analysis presented here are a step forward on the path to precision epidemiology.

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

## Methods

For all Methods subsections, greater detail is provided in the Supplementary Methods.

### Data

All data for this study were derived from contacts notified by the NHS COVID-19 contact-tracing app between April 2021 and February 2022 inclusive. The data-generating process for app data was non-trivial: the primary aim was successful implementation of a privacy-preserving and data-minimizing contact-tracing process, not generating data for epidemiological study. We analysed data recorded by the app with three different timings/frequencies: (1) daily 'analytics' data, (2) exposure data sent when a contact is notified of risky exposure and (3) exposure data sent when a contact reports a positive test. Nowhere in the data is there a unique identifier for each app user, and so connecting these three data sources required some application of logic, some assumption and some subsetting of the data. We next explain each of these three data sources in turn.

First, we previously described the daily analytics data[6,7]. Each correctly functioning installation of the app sent one 'analytics packet' of data daily (at midnight, regardless of whether the user was notified that day). Each packet indicated whether or not the app user had been notified of risky exposure on that day and included four fields of 'individual characteristics' that we assumed were usually constant for an individual over the time scale of one round of contact tracing and testing (that is, are effectively constant for the individual): their device model (for example, iPhone X), their operating system version on this device, the postcode district (an area with mean population size of about 20,000 individuals) in which they reported residing and their lower-tier local authority (if ambiguous from the postcode district).

Second, when a contact was notified of a risky exposure to an anonymous case, their app sent one 'event packet' of data to the central server for each exposure window (lasting a maximum of 30 min) that had a risk score over the threshold for notification. These were sent separately from the daily analytics packets, and only at the time of notification. Data relating to proximity to any individual not reporting a positive test are never sent to the central server. Event packets included information on exposure proximity, duration and date and the same four fields of individual characteristics as in the daily analytics packets. Events packets contained no information about the index case to whom the contact was exposed (such information is irretrievable by the app by design), except for whether their infectiousness at the time of exposure was encoded as high or standard. If a single, continuous exposure event lasted more than 30 min it was automatically split into multiple exposure windows that were considered separately; multiple exposures occurring at different times (that is, a discontinuous meeting between the individuals) also resulted in separate exposure windows. Risk calculations were performed separately on each exposure window. As explained above, the overall risk score used by the app for each window was calculated by multiplying scores from proximity, duration and index infectiousness, and we normalized these overall scores by the value for a 15-min exposure to an index case of standard infectiousness at a proximity of 2 m. With this normalization the threshold for notification used by the app was a risk score of 1.11 throughout the period analysed; this value was chosen as part of the intervention deployment, not as part of the analysis here.

Third, if an individual reported a positive test in the app during the 'observation interval'—starting with their notification and ending 14 days after the exposure—the same event packets that were sent when the individual was notified were sent once more to the central server, identical except for a flag indicating that this was the report-positive stage rather than the notification stage.

Jointly analysing the second and third data sources—the event packets sent at notification and again at positive test—we could assign to each exposure window the binary outcome of 'positive test reported or not'. This follows because we could see which event packets were sent a second time with all data fields identical except for the flag indicating either notification or report-positive stage, and which event packets were not. An assignment of a reported-positive-test outcome to a given exposure window does not imply that that exposure window was causal for the individual becoming infected: the transmission event could have been caused by background risk or by any other exposure window for the same contact if they had multiple exposure windows.

When more than one risky exposure window was recorded between a contact and the index case, these were analysed separately for the risk calculation and sent as separate event packets to the central server. The absence of a unique individual identifier means that in general one cannot know whether $n$ event packets sent on the same day (as determined by the date received centrally) with matching individual characteristics for the contact (device model, operating system version, postcode district and lower-tier local authority) were sent by (1) one contact with $n$ risky exposure windows, (2) $n$ contacts who were notified on the same day and had matching individual characteristics, with one risky exposure window each, or (3) anything in between. We therefore restricted the dataset of event packets to an unambiguous subset constructed as follows. From the daily analytics data we identified the subset of notifications (of risky exposure) when exactly one contact with a given combination of individual characteristics was notified on a given day; for each such notification we assumed that all event packets with identical characteristics originated from the same contact—scenario (1) above. When more than one contact with given characteristics was notified on a given day, all event packets that day with those characteristics were excluded from analysis for simplicity. This procedure for grouping multiple event packets as being from the same contact is specifically for a single notification event of a given contact: if the same individual is notified multiple times during our study, each notification event (which will be at least a quarantine period apart from other notifications, by design) is treated as being from a separate individual, with a set of event packets associated with each event.

Extended Data Table 1 summarizes sample sizes for the final dataset analysed in this paper. Supplementary Table 1 summarizes sample sizes and aspects of the events packet data at three of the stages described above: before and after the grouping stage, and also for only those contacts who reported a positive test. The grouping stage—subsetting to instances when only a single contact with given characteristics was notified on a given day, for which the matching event packets can be grouped as from one contact—retains 60% of the events packets.

### Empirical estimation of individuals' probability of testing positive from summary statistics

In general, each contact in our dataset had multiple exposure windows, each of which had a duration (anything up to 30 min) and a risk score. We summarized these data for each contact into metrics including the maximum risk score from any of the windows, the cumulative risk score over all windows and the cumulative duration over all windows. We binned (grouped) contacts by the value of their summary metrics and, within each bin, calculated the fraction of contacts reporting a positive test in the observation interval. Confidence intervals on this fraction were calculated through the associated binomial distribution (defined with the number of 'trials' equal to the group size and the number of 'successes' equal to the number of contacts reporting a positive test). We extrapolated our estimates to risk score 1 (that is, 2 m away from an index case with standard infectiousness for 15 min (indicated by a grey circle in Fig. 1) as a point of comparison) via a quadratic fit. In Figs. 2 and 4 the background risk estimate from the maximum-likelihood approach outlined below was subtracted from the result. In all figures

the $x$ coordinate for each bin corresponds to the mean of all scores within the bin.

## Statistical modelling of the per-exposure-window probability of transmission

In reality, a given individual who reported a positive test was infected either by the background, or in their first recorded window, or in their second recorded window, and so on, but which of these was actually the case is unknown. Hence we modelled the process in terms of risk parameters, shared between individuals, that are to be estimated. We developed a statistical model for the separate contributions to each individual's overall risk from each of their exposure windows and from background risk. Specifically, we modelled the probability of individual $i$ not reporting a positive test during the observation interval as

$$(1 - B_i) \times (1 - P_t(i\text{'s first window})) \times (1 - P_t(i\text{'s second window})) \times \ldots$$
$$\times (1 - P_t(i\text{'s last window}))$$

where $B_i$ is the probability of background transmission (followed by reporting of a positive test) and $P_t$ ($i$'s $n$th window) is the probability of transmission during the $i$'s $n$th window (followed by reporting of a positive test). The justification for this form is that if an individual does not report a positive test, this implies that they were not infected by the background (with subsequent reporting) and were not infected during their first window (with subsequent reporting) and not during their second window, and so on. The probabilities for each of these events not happening should thus be multiplied to give the overall probability for none of them happening. We modelled $B_i$ as $1 - (1 - b_i)^{\beta}$, defining $b_i$ as the sum, over the 14 days following $i$'s notification, of the weekly smoothed mean daily fraction of geographically matched, not-recently-notified app users that reported a positive test ($\beta$ is the associated regression coefficient for this term). For small values of $b_i$ the background risk is simply rescaled by factor $\beta$—that is, $B_i \approx \beta b_i$; for larger values of $b_i$ the functional form accounts for saturation of risk. We modelled $P_t$ ($i$'s $n$th window) as depending only on the risk score recorded by the app for $i$'s $n$th window. We binned risk scores into eight bins, defining a single independent $P_t$ parameter for each bin, such that the expression above could be rewritten as

$$(1 - B_i) \times \prod_{j=1}^{8} (1 - P_t(\text{bin } j))^{(\text{number of windows from } i \text{ with risk score in bin } j)}.$$

The probability that individual $i$ would report a positive test during the observation interval is 1 minus the expression above (the expression for them not reporting a positive test in the interval). The likelihood is given by the product of all individuals' probabilities for their reported outcome for testing positive. We maximized the likelihood to estimate the parameters $\beta$ and per-window transmission probability for each of the eight bins of risk score (plotted in Fig. 3) and profiled the likelihood of obtaining the confidence intervals. Figure 3 shows that the per-window transmission risk estimated for each of the eight bins is proportional to the app-recorded risk score of that bin. We used a binning approach to allow the data to show this proportionality—rather than taking it to be true as a modelling assumption—because this proportionality serves as validation for the app's risk score capturing real risk.

As a robustness check we developed likelihoods based on frailty models with several sources of heterogeneity among case–contact pairs in the model (Supplementary Methods section 1.6.2).

## Predictors and machine learning classifiers

As basic input predictors for machine learning we used maximum, mean and cumulative risk scores and the duration and number of exposures in each bin of risk score. Additional predictors included date, region, rural/urban score, background rate of infections, day of the week with more exposure windows and peak daily duration. Classifiers used included logistic regression, gradient-boosting machines[31] and extreme gradient-boosting XGBoost[32] with 10, 100 and 400 rounds.

Optimal strategies for amber notifications were obtained using a general approach for targeted interventions[33] presented in Supplementary Discussion.

## Reporting summary

Further information on research design is available in the Nature Portfolio Reporting Summary linked to this article.

## Data availability

Data access is managed by UKHSA, who will make available on request the data needed to replicate the key results, either via the UK Data Service or through direct request for data access to UKHSA (details on the process can be found at https://www.gov.uk/government/publications/accessing-ukhsa-protected-data). Access is controlled for privacy reasons.

## Code availability

The R code for the analyses in this paper is available on the official UKHSA repository at https://github.com/ukhsa-collaboration/risk_scoring_nhs_covid19_app.

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

**Acknowledgements** We thank teams across the UK Health Security Agency (UKHSA) and previously at NHS Test and Trace for help and support. In particular we thank the NHS COVID-19 app Data and Analytics Team for their invaluable support with data access, management and analytics. This work was funded by a Li Ka Shing Foundation award and research grant funding from the UK Department of Health and Social Care (DHSC), both to C.F., and by the National Institute for Health and Care Research to the Health Protection Research Units in Genomics and Enabling Data (grant no. NIHR200892 to M.K.) and in Healthcare Associated Infections and Antimicrobial Resistance (grant no. NIHR200915, a partnership between UKHSA and the University of Oxford). The views expressed in this article are those of the author(s) and are not necessarily those of UKHSA or DHSC.

**Author contributions** L.F., C.W., J.P., A.L., M.C., M.B. and C.F. conceptualized this work. L.F., C.W., J.P., A.F., M.K., D.T. and C.F. performed analyses. All authors contributed to the writing and reviewing of the manuscript.

**Competing interests** L.F., C.W. and C.F. were named researchers on a grant from DHSC to Oxford University. M.K. has a data-sharing agreement with UKHSA. D.T. was an employee of Zühlke, which provided consultancy to UKHSA. M.C., M.B., A.L., J.P.-G. and A.D.F. were employees of, or were affiliated to, UKHSA.

**Additional information**
**Correspondence and requests for materials** should be addressed to Luca Ferretti or Christophe Fraser.

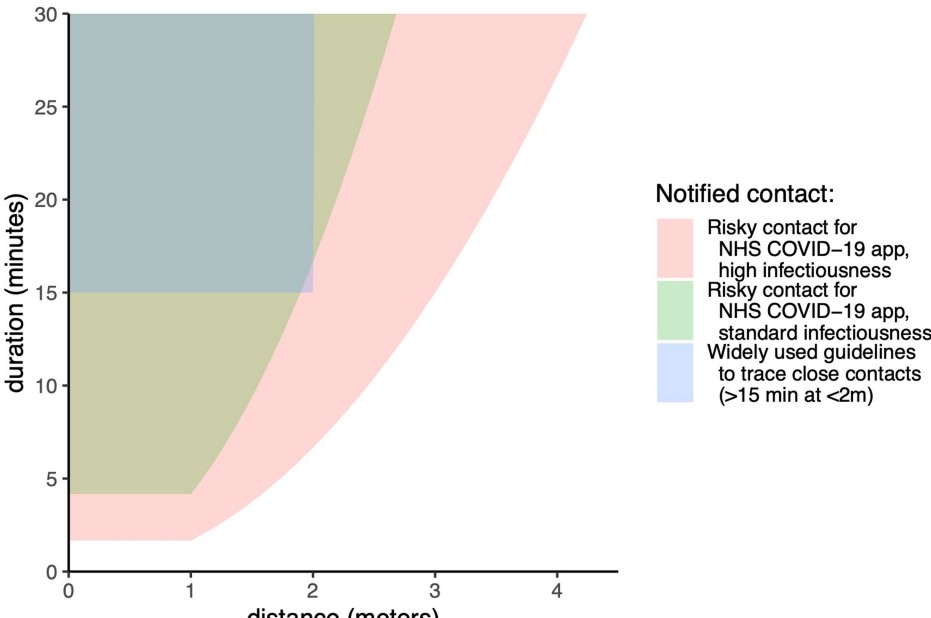

**Extended Data Fig. 1 | The app has more nuanced distance-duration rules than manual contact tracing.** Coloured regions show regions of the distance-duration space where contacts are notified digitally (depending on the infectiousness of the index case) or manually. These boundaries apply in theory, though in practice distances are imperfectly estimated from Bluetooth signal attenuation.

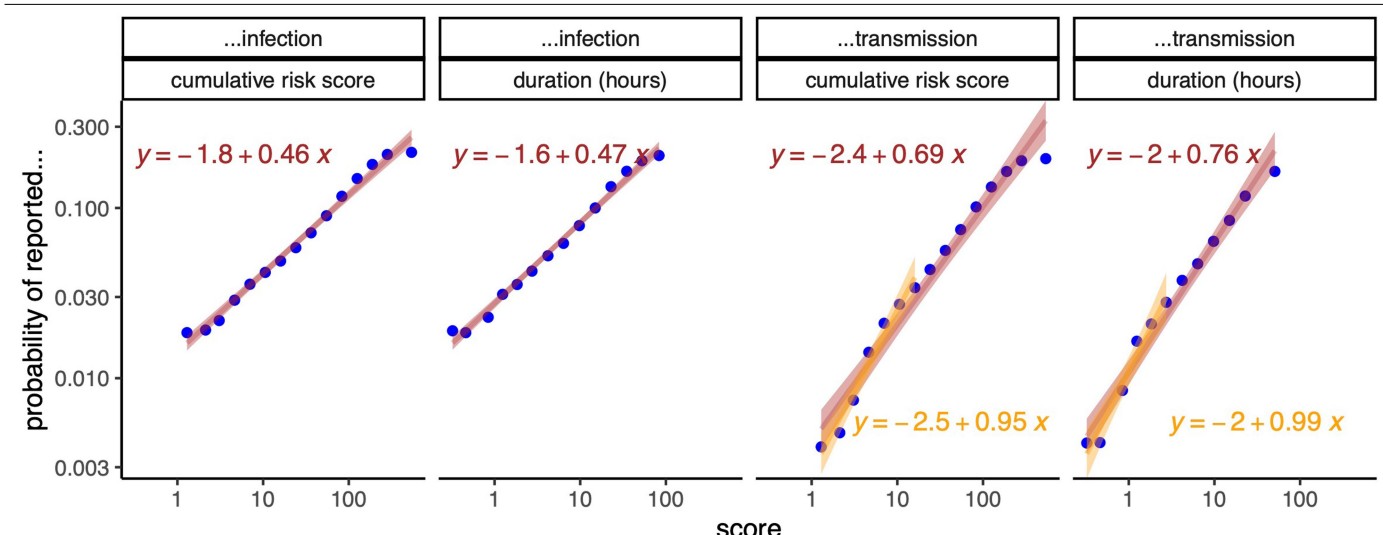

**Extended Data Fig. 2 | The probability of transmission depends linearly on duration and cumulative risk for short exposures, then sublinearly.** Log-log plots of the probability of reported infection (the fraction of notified contacts who report a positive test shortly after notification) and transmission (subtracting the maximum-likelihood correction for background risk) as a function of cumulative risk score and duration of exposure. Points correspond to maximum likelihood estimates. The brown bands show the 95% confidence interval for linear regressions on the points shown, i.e. a power-law relation between risk predictors and the probability of reporting a positive test. The maximum-likelihood estimates for the exponents are $P_t \sim r_{cum}^{0.46\pm0.01}$, $P_t \sim d^{0.47\pm0.01}$ (infection) and $P_t \sim r_{cum}^{0.69\pm0.04}$, $P_t \sim d^{0.76\pm0.04}$ (transmission). For the regressions of the probability of transmission, when restricting to low values of the risk predictor (cumulative risk <20, duration <3 h), the relationships were approximately linear: $P_t \sim r_{cum}^{0.95\pm0.07}$, $P_t \sim d^{0.99\pm0.09}$ (orange bands), as expected from theoretical arguments. The ± values shown in the exponents are standard deviations.

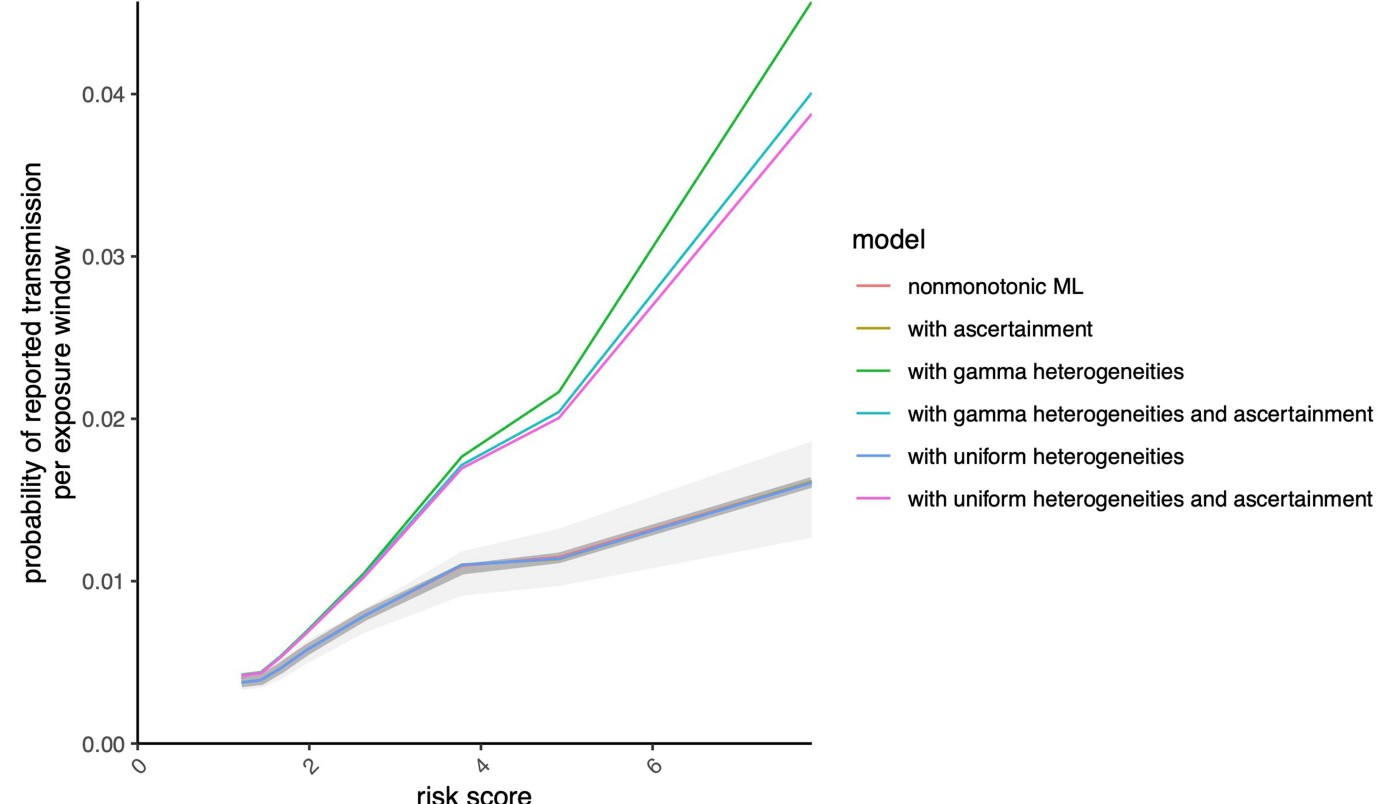

**Extended Data Fig. 3 | The monotonic relationship between the risk score per window and the probability of transmission in that window is robust with respect to the inclusion of individual heterogeneities in the model.** Maximum-likelihood estimates of the probability of reported transmission per exposure window, i.e. the estimated probability of transmission in an individual exposure window followed by reporting of a positive test, as a function of the binned app-measured risk score for that window. The grey line and shading show the maximum-likelihood monotonic risk (and the corresponding 95% CI) shown in Fig. 3. Lines of different colours show maximum-likelihood estimates from models that do not assume monotonicity; these models include positive-test ascertainment and/or different functional forms for heterogeneities in risk (see Supplementary Methods Section 1.6.2).

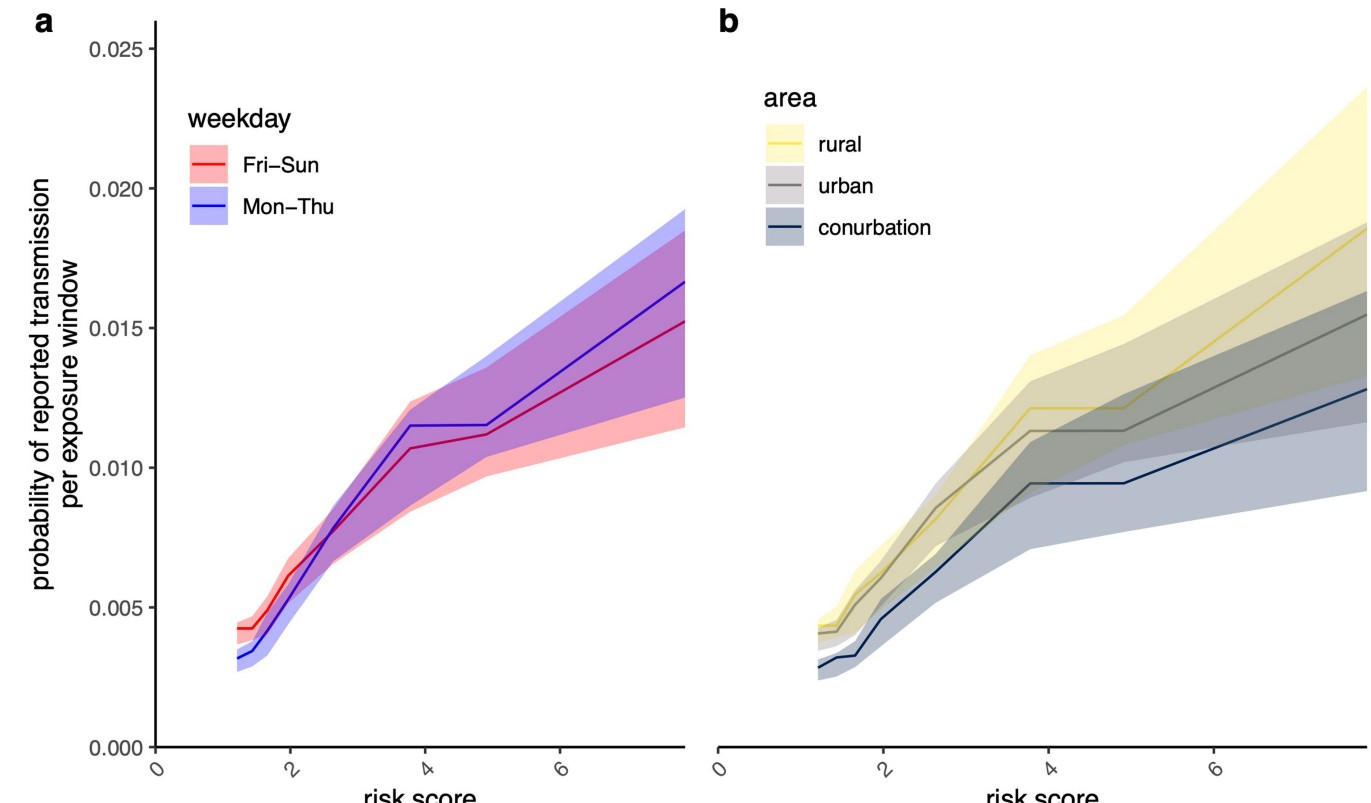

**Extended Data Fig. 4 | The transmission probability per exposure window decreases for contacts located in conurbations and increases for low-risk exposures during the weekend.** The probability of reported transmission per exposure window, i.e. the estimated probability of transmission in an individual 30-minute exposure window followed by reporting of a positive test, is shown as a function of the app-measured risk score for that window, as in Fig. 3 but with stratifications of contacts. Panel a: Stratification by weekday or weekend. Panel b: Stratification by rural area, urban area (town or city) and conurbation (urban agglomeration). Lines connect the maximum-likelihood estimates for each bin; shaded areas indicate 95% confidence intervals.

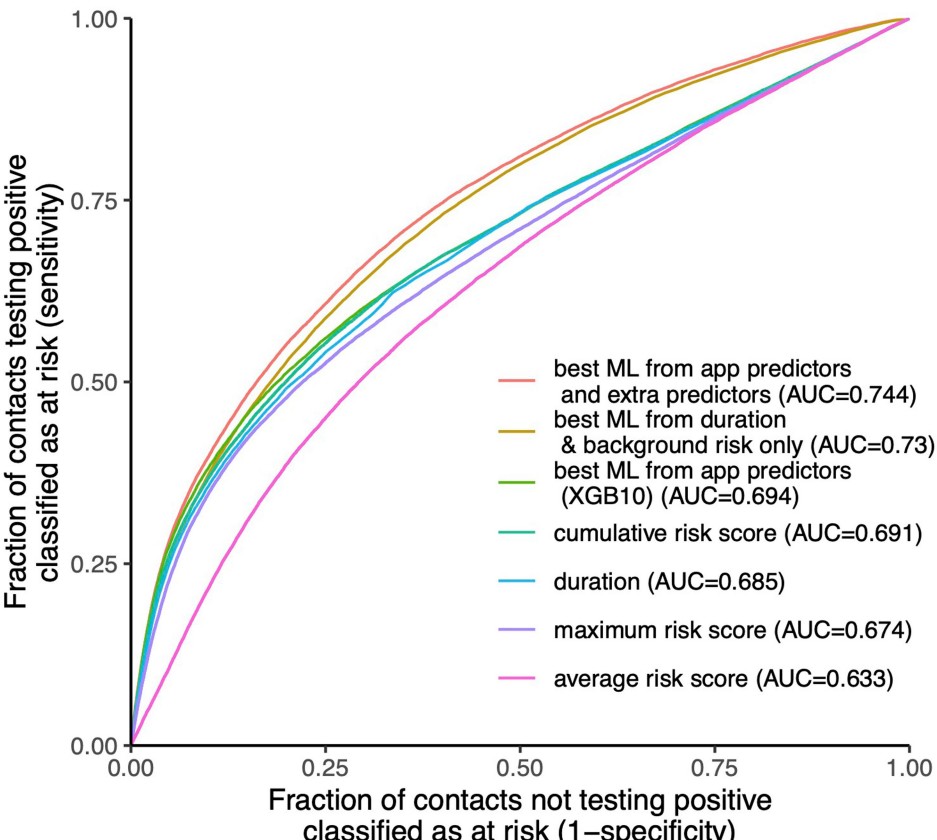

**Extended Data Fig. 5 | Duration and cumulative risk are the best predictors of infection, only marginally improved by machine learning.** Sensitivity/specificity (receiver operating characteristic) curve for different methods and thresholds to classify individuals exposed to an index case as at risk or not. Our dataset contained only individuals who were actually notified; we varied the classification thresholds to interpolate between continuing to notify all of these individuals (top right) and notifying none of these individuals (bottom left). Different colours show different classification methods. For each method we varied thresholds to explore their balance between sensitivity (notifying individuals who would report a subsequent positive test) and specificity (not notifying individuals who would not). ML abbreviates machine learning, AUC the area under the curve.

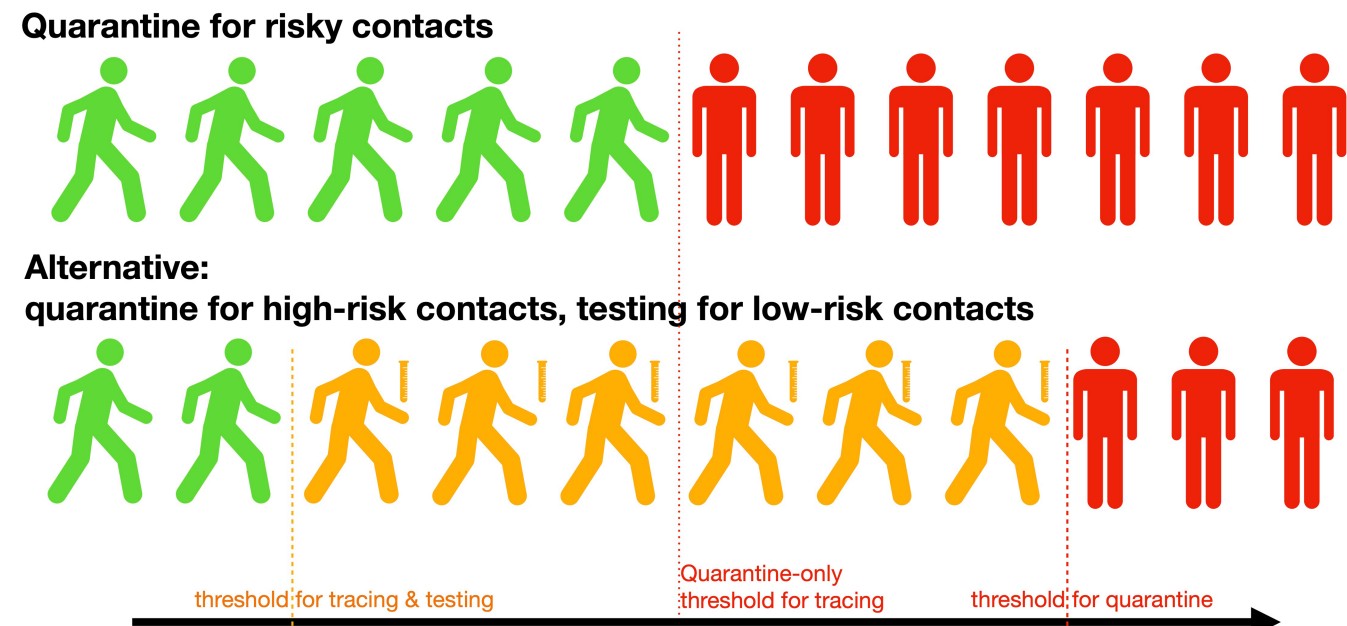

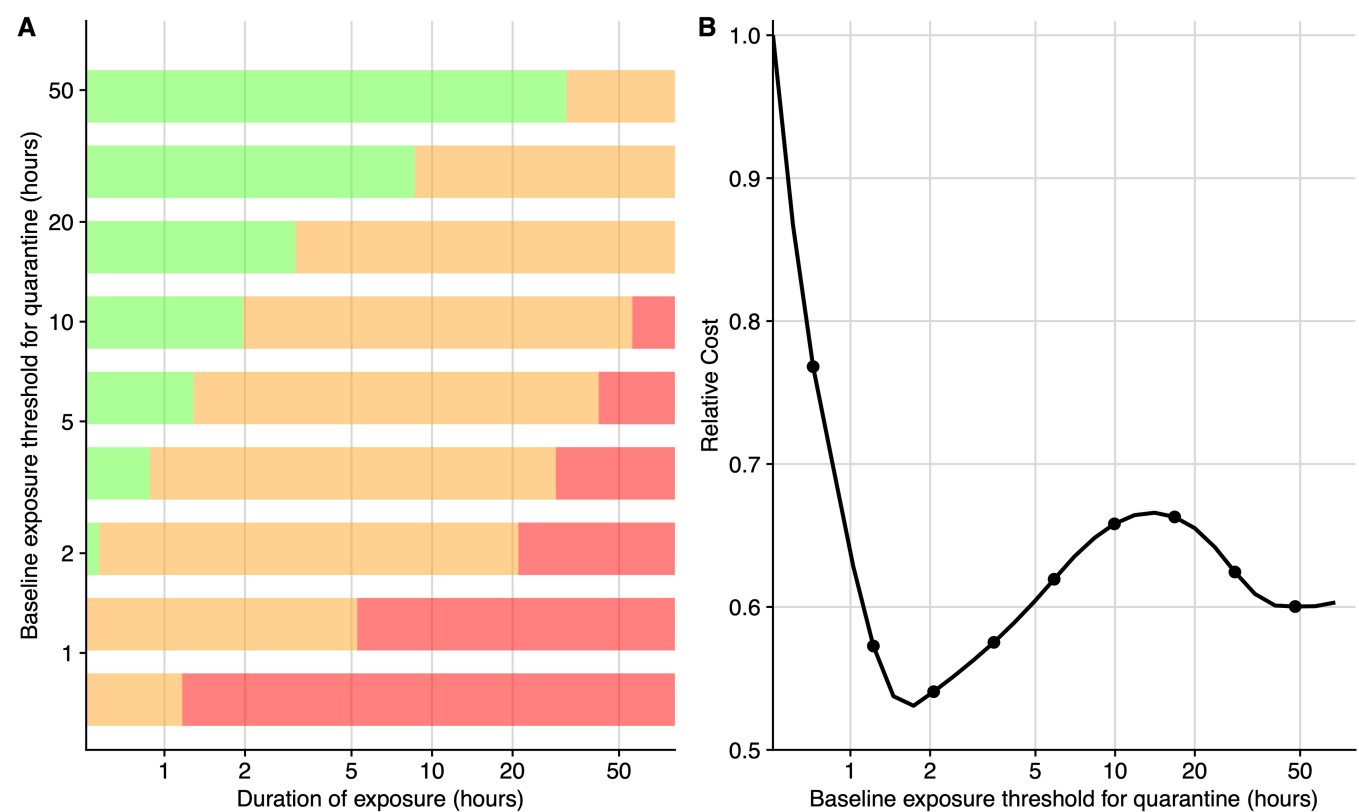

**Extended Data Fig. 6 | Illustration of optimal strategies to reduce social costs of contact tracing via amber/red alert notifications.** In this illustrative scenario we considered combinations of three measures: red notification leading to quarantine after notification, amber notification leading to PCR test after notification (followed by self-isolation if positive), and no notification. We assume that the risk of infection would be assessed based on duration of exposure. We consider optimal strategies leading to minimisation of total costs for patient and public health for a given epidemiological effectiveness; see Supplementary Discussion for details and assumptions on relative costs and effectiveness. Panel a: each horizontal line represents an optimal strategy (quarantining high-risk contacts, testing intermediate-risk contacts, not tracing low-risk contacts) that has the same effectiveness as a baseline quarantine-only strategy for contacts above a threshold duration of exposure (y axis). Panel b: the decrease in cost of the optimal strategy relative to the baseline strategy (quarantine for all traced contacts).

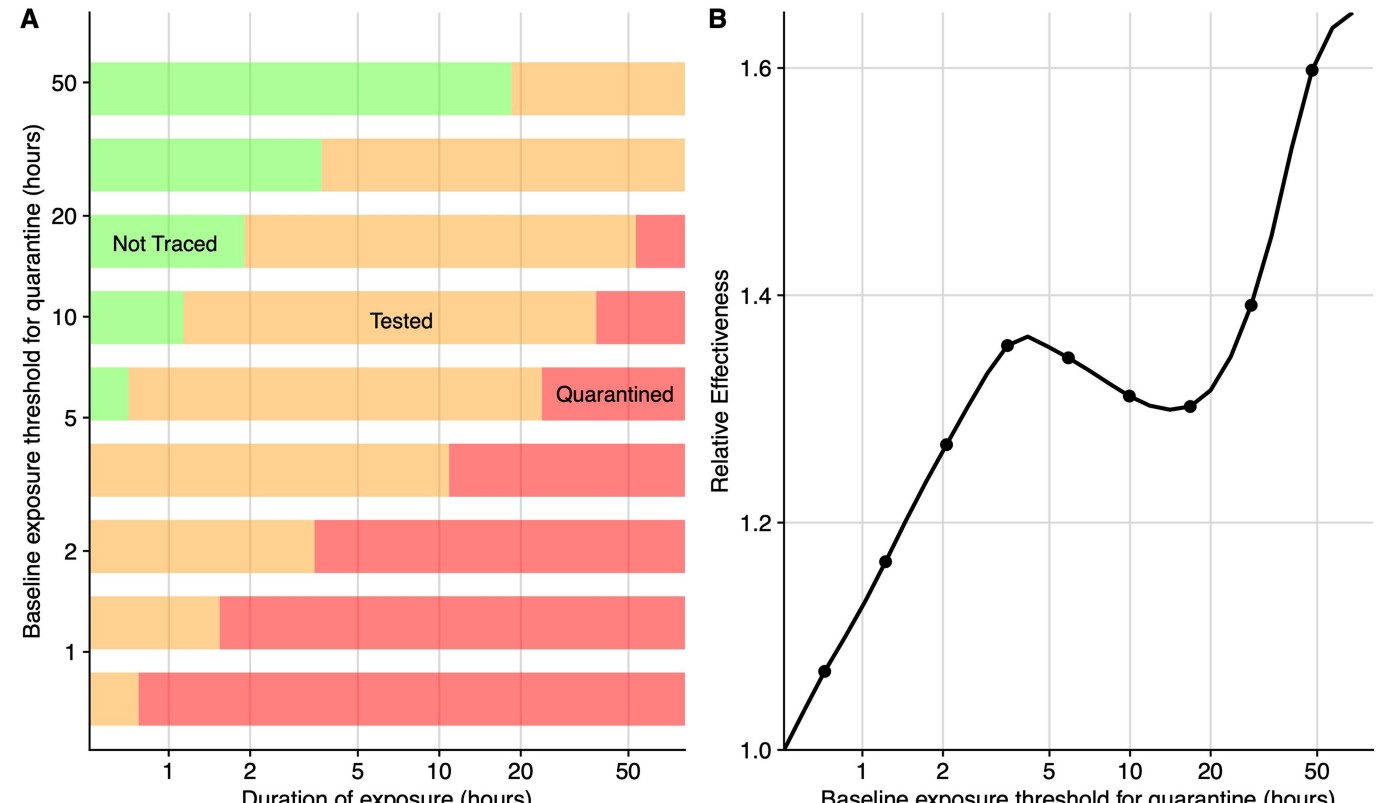

**Extended Data Fig. 7 | Illustration of optimal strategies to increase effectiveness of contact tracing via amber/red alert notifications.** Same as Extended Data Fig. 6, but considering optimal strategies that keep the total costs fixed while maximising epidemiological effectiveness. Panel a: each horizontal line represents an optimal strategy that has the same cost as a baseline quarantine-only strategy for contacts above a threshold duration of exposure (y axis). Panel b: the increase in effectiveness of the optimal strategy relative to the baseline strategy.

**Extended Data Table 1 | Summary statistics for the NHS COVID-19 app exposure dataset**

| | Grouped events packets | Grouped event packets for individuals reporting a positive test |
|---|---|---|
| **Number of packets** | 52,979,100 | 6,137,896 |
| **Cumulative duration (hours)** | 22,877,575 | 2,670,167 |
| **Mean duration per window (minutes)** | 26 | 26 |
| **Number of contacts** | 7,047,541 | 239,683 |
| **Mean duration per contact** | 3 hours and 15 minutes | 11 hours and 8 minutes |
| **Mean number of packets per contact** | 7.52 | 25.61 |

We report statistics only for exposure windows that were successfully grouped and assigned to a single contact. These windows represent about 60% of the whole dataset. See Supplementary Table S1 for further details on the raw exposure window data before the grouping stage.

**Extended Data Table 2 | Summary statistics for different types of contacts in our dataset**

| Type of contact | mean duration (hours) | mean risk score | % testing positive | mean risk score per hour | % of all contacts | % of duration from all exposures | % of cumulative risk score from all exposures | % of all transmissions |
|---|---|---|---|---|---|---|---|---|
| *household* | *32.9* | *173.9* | *13* | *5.3* | *6* | *61* | *67* | *41* |
| *recurring* | *3.9* | *16.6* | *3.3* | *4.2* | *14* | *17* | *15* | *24* |
| *single day* | *1.6* | *6.6* | *1.5* | *4.2* | *32* | *15* | *13* | *25* |
| *fleeting* | *0.4* | *1.5* | *0.4* | *3.7* | *48* | *6* | *4.5* | *10* |

Household contacts (defined as contacts whose exposures cover more than 15 windows in a single day), recurring contacts (defined as non-household contacts whose multiple exposure windows occur on two different days or more), one-day contacts (defined as non-household contacts whose multiple exposure windows occur all in a single day) and fleeting contacts (defined as contacts with a single exposure window).

# Reporting Summary

## Statistics

For all statistical analyses, confirm that the following items are present in the figure legend, table legend, main text, or Methods section.

| n/a | Confirmed | |
|---|---|---|
| ☐ | ☒ | The exact sample size (*n*) for each experimental group/condition, given as a discrete number and unit of measurement |
| ☒ | ☐ | A statement on whether measurements were taken from distinct samples or whether the same sample was measured repeatedly |
| ☐ | ☒ | The statistical test(s) used AND whether they are one- or two-sided *Only common tests should be described solely by name; describe more complex techniques in the Methods section.* |
| ☒ | ☐ | A description of all covariates tested |
| ☐ | ☒ | A description of any assumptions or corrections, such as tests of normality and adjustment for multiple comparisons |
| ☐ | ☒ | A full description of the statistical parameters including central tendency (e.g. means) or other basic estimates (e.g. regression coefficient) AND variation (e.g. standard deviation) or associated estimates of uncertainty (e.g. confidence intervals) |
| ☒ | ☐ | For null hypothesis testing, the test statistic (e.g. *F*, *t*, *r*) with confidence intervals, effect sizes, degrees of freedom and *P* value noted *Give P values as exact values whenever suitable.* |
| ☒ | ☐ | For Bayesian analysis, information on the choice of priors and Markov chain Monte Carlo settings |
| ☒ | ☐ | For hierarchical and complex designs, identification of the appropriate level for tests and full reporting of outcomes |
| ☐ | ☒ | Estimates of effect sizes (e.g. Cohen's *d*, Pearson's *r*), indicating how they were calculated |

*Our web collection on statistics for biologists contains articles on many of the points above.*

## Software and code

Policy information about availability of computer code

| Data collection | Data collection was performed using RAthena (v2.6.1) queries of the database of private app data. |
|---|---|
| Data analysis | Analysis was performed in R, version 4.0.4, with use of packages data.table (v1.14.2), tidyverse (v1.3.2), gbm (v2.1.8.1), xgboost (v1.6). Code to replicate the analysis will be made available as part of the data sharing process by UKHSA at https://github.com/ukhsa-collaboration/risk_scoring_nhs_covid19_app. |

For manuscripts utilizing custom algorithms or software that are central to the research but not yet described in published literature, software must be made available to editors and reviewers. We strongly encourage code deposition in a community repository (e.g. GitHub). See the Nature Portfolio guidelines for submitting code & software for further information.

## Data

Policy information about availability of data

All manuscripts must include a data availability statement. This statement should provide the following information, where applicable:
- Accession codes, unique identifiers, or web links for publicly available datasets
- A description of any restrictions on data availability
- For clinical datasets or third party data, please ensure that the statement adheres to our policy

Data access is managed by UKHSA, who will make available on request the data needed to replicate the key results, either via the UK Data Service or through direct

request for data access to UKHSA (details on the process can be found at https://www.gov.uk/government/publications/accessing-ukhsa-protected-data). Access is controlled for privacy reasons.

# Research involving human participants, their data, or biological material

Policy information about studies with human participants or human data. See also policy information about sex, gender (identity/presentation), and sexual orientation and race, ethnicity and racism.

| | |
|---|---|
| Reporting on sex and gender | N/A |
| Reporting on race, ethnicity, or other socially relevant groupings | N/A |
| Population characteristics | N/A |
| Recruitment | N/A |
| Ethics oversight | Research Ethics Committee approval was not required because our analysis was performed on routinely collected, anonymised data that cannot be traced back to individuals, from a database built with the primary purpose of supporting public health. |

Note that full information on the approval of the study protocol must also be provided in the manuscript.

# Field-specific reporting

Please select the one below that is the best fit for your research. If you are not sure, read the appropriate sections before making your selection.

☒ Life sciences        ☐ Behavioural & social sciences        ☐ Ecological, evolutionary & environmental sciences

For a reference copy of the document with all sections, see nature.com/documents/nr-reporting-summary-flat.pdf

# Life sciences study design

All studies must disclose on these points even when the disclosure is negative.

| | |
|---|---|
| Sample size | We used all available data |
| Data exclusions | No data were excluded from the analyses |
| Replication | This is a one-off observational study, with no replication possible. |
| Randomization | Not relevant - no use of experimental groups |
| Blinding | Not relevant - no use of experimental groups |

# Reporting for specific materials, systems and methods

We require information from authors about some types of materials, experimental systems and methods used in many studies. Here, indicate whether each material, system or method listed is relevant to your study. If you are not sure if a list item applies to your research, read the appropriate section before selecting a response.

## Materials & experimental systems

| n/a | Involved in the study |
|---|---|
| ☒ ☐ | Antibodies |
| ☒ ☐ | Eukaryotic cell lines |
| ☒ ☐ | Palaeontology and archaeology |
| ☒ ☐ | Animals and other organisms |
| ☒ ☐ | Clinical data |
| ☒ ☐ | Dual use research of concern |
| ☒ ☐ | Plants |

## Methods

| n/a | Involved in the study |
|---|---|
| ☒ ☐ | ChIP-seq |
| ☒ ☐ | Flow cytometry |
| ☒ ☐ | MRI-based neuroimaging |

