## [Peer Review File · Nature]

Manuscript Title: Digital measurement of SARS-CoV-2 transmission risk from 7 million contacts

Reviewer Comments & Author Rebuttals

Reviewer Reports on the Initial Version:

Referees' comments:

Referee #1 (Remarks to the Author):

Thank you for the opportunity to review the study by Ferretti and colleagues. In this article, the authors aim to demonstrate how risk scores derived from proximity tracing apps are associated with the risk of a contact person testing positive for SARS-CoV-2 within the next 14 days. This risk score integrates three components, including the proximity between two apps, the duration of proximity contact, and the likely infectiousness of the infected case (defined by the timing of contact relative to the symptom onset of the case). The authors report several noteworthy findings. First, the probability of exposure notified contacts to test positive for SARS-CoV-2 indeed increases with various risk score indicators or proximity contact duration. A more detailed analysis further identifies the length of contact as the probably most important risk component for transmission. The study further reports a disproportionately high contribution of same-household contacts (with intrinsically longer proximity contact durations). The study concludes that with further refinements of proximity tracing (e.g. by adding an "amber" warning level) and by adapting recommended measures upon exposure notification according to risk score levels, digital proximity tracing could become an important tool towards more precise (i.e. personalized) public health measures for transmission mitigation.

This is a very important study, and I applaud the authors for undertaking this detailed and thorough analysis. The study sheds further light on possible future applications of digital proximity tracing. So far, the effectiveness of such apps was mainly centered around warning exposed contacts earlier, more widely (including exposures that may be missed by conventional contact tracing), and in a more scalable fashion. The present analysis suggests that digital proximity tracing also enables more precise public health measures. Instead of recommending immediate quarantine for all exposed contacts, digital proximity tracing enables more nuanced public health measures (e.g. "test and wait") by leveraging risk scores. Therefore, the article suggests a way forward for making digital proximity tracing even more useful in future epidemics.

Several aspects warrant closer inspection, however.

Data filtering and exposure window linkage/matching

=====

First and foremost, the authors search and match a large database of exposure notifications (windows) and reported positive tests. I strongly recommend that the authors present the filtering

steps more concisely, provide more actual numbers (sample sizes), and discuss the limitations and potential biases more comprehensively. The sample sizes/numbers seem to be dispersed in multiple places in the main document and supplements. It would be helpful to have them all in one place, e.g. in a flow chart. Furthermore, it is unclear how the various filtering and matching procedures affect the generalizability of the main results. For example, the authors select around 53 Mio. exposure notifications (ENs). In the next step, they try to assign the ENs that belong to an "exposure window" (i.e. sequence of continuous 30-exposure windows) by a specific contact based on a probabilistic linkage. The linkage occurs based on information regarding exposure dates, iOS, mobile phone type, and zip codes provided voluntarily (in only around 20K participants). The authors claim that they have been able to uniquely assign 60% (footnote table 1) of ENs to specific exposure windows. In other words, 40% could not be assigned unambiguously, which is a substantial number. The filtering and matching steps seem quite crucial for the analysis. Hence, can the authors please provide actual numbers for matches, non-matches, and context (e.g. exposure date)? It seems to me that matching on exposure date, device, and OS alone will create large numbers of ambiguous matches - especially in high-incidence phases. Did the authors notice specific temporal patterns or associations with the incidence in "linkage success" (I am unsure whether this is what the authors modeled in subsection 1.5.2. in the supplementary materials)?

Along similar lines, the authors report that around 200K (out of 7 Mio) exposure contacts reported follow-up positive test results within the likely exposure window. Again, the selection and matching of exposure windows matter here because these processes influence probability calculations for testing positive after exposure. My questions here: What was the denominator for the probability of infection analysis? Was it the 7 Mio contacts - assuming that those not reporting a positive result never tested positive during the exposure window? A positive test may be missing for multiple reasons: no test was done, the test was negative, the test was positive but not reported, or the test was reported but the information was not transmitted due to technical errors. Several assumptions surrounding the accuracy of positive tests are reported in the supplementary materials (e.g. lines 104-131 or section 1.7). But they do not appear to investigate (or I fail to see it) whether the underlying assumption that the absence of positive tests means test negativity (in a mathematical sense) is justified.

Furthermore, the authors note a tendency for long exposure durations (which is shown in Extended Table 1 and Figures 4). What makes the authors confident that this is truly a valid finding and not an artifact of their filtering and linkage procedures? In other words, is there supporting evidence that 1) the 40% excluded exposure notifications are not different (e.g. in terms of exposure duration or risk scores) compared with the 60% included (btw. the 60% stems from the footnote of ET1), and 2) the authors also seem to assume that the 200K persons reporting positive test results are comparable to contacts who tested positive but chose not to report their positive test result in the app.

Overall, the authors should discuss more clearly how filtering and linkage may have introduced biases and/or affected the robustness of their observations, particularly concerning their main finding of infection probability mainly depending on exposure duration.

Technical differences between the NHS Covid-19 app and fully decentralized GAEN apps

=====

This may be a bit picky, but in the Introduction of the Supplementary Materials the UK NHS Covid-19 app is portrayed as relying on the Google Apple Exposure Notification (GAEN) framework. While this may technically be true (I am unable to say so), the UK app deviates from other GAEN apps in important ways. Specifically, the UK app stores exposure notifications in a central server, which enables the matching of exposure notifications with positive tests. Most apps I am familiar with have an entirely decentralized architecture, where exposure notifications are entirely analyzed on mobile phones, and no data is sent back to centralized servers.

The architectural differences are relevant for the discussion of future directions of leveraging digital proximity tracing for more precise public health actions. The final sentence and recommendation (line 429, as well as line 356ff.) to integrate risk scores into public health toolboxes may be quite controversial in some settings. I remember quite heated debates with international colleagues about whether and how digital proximity tracing apps should also be turned into data collection tools to learn more about transmission dynamics (as proposed in line 401). I think it may be important to acknowledge that some jurisdictions may not want to collect exposure notifications in a centralized system. Thus, how would the author's findings still be useful to make contact tracing more precise in systems and/or in settings with fully decentralized digital proximity tracing? The only way I could see this work if risk scores and alert levels are only stored locally on mobile phones, is if contact tracers ask app users to send a screenshot of alert levels (amber or red) or the like.

Overemphasis of the word "precision" in some contexts (in my opinion)

=====

I very much agree with the aim of precision public health, and the author's work is a really important milestone to make contact tracing *more precise*. However, I have some issues with the claim that using risk scores equates to precision contact tracing (line 55, line 401 which speaks about high-precision risk assessment) or line 35 in the summary. But it is still a very long way to real precision - as the authors acknowledge themselves in lines 403ff.

The role of aerosol transmission

=====

The initially proposed heuristic of 2m /15min for contact tracing, as well as distancing rules for protection, were based on the assumption that SARS-CoV-2 transmission would mainly occur via droplets. Meanwhile, there is also substantial evidence for aerosol transmission of SARS-CoV-2 (e.g. <https://www.bmj.com/content/377/bmj-2021-068743>, [https://www.thelancet.com/journals/lancet/article/PIIS0140-6736\(21\)00869-2/fulltext?ref=vc.ru](https://www.thelancet.com/journals/lancet/article/PIIS0140-6736(21)00869-2/fulltext?ref=vc.ru)). Against this backdrop, can the authors speculate whether and how aerosol transmission (as opposed to "proximity-based" risks) may have influenced their findings and their generalizability? It seems to me that aerosol transmission would not be captured by this analysis unless some relevant proximity (leading to a risk score above the threshold) was also involved. For instance, could the author's finding of substantial infection risks for "longer exposures at greater distances" (line 28f, line 365) reflect aerosol transmission of SARS-CoV-2? On another note, the inability of digital proximity tracing

apps to accurately measure aerosol transmission is yet another reason why I would refrain from calling risk score-augmented manual contact tracing "high precision contact tracing".

Minor comments

Line 47: Can you add references for the "social and economic costs" of conventional epidemic control measures?

Lines 54 and 133: Examples where I think the "precision" aspect should be toned down a bit, e.g. "approach towards precision epidemiology" in line 54

Line 57: digitized the process of contact tracing. I am sure persons involved in manual contact tracing would object to this statement. Manual contact tracing assesses more information than "just" proximity.

Figure F1: please use consistent labels, i.e. risk scores rather than risk predictor. The second panel from left, top row: values of x-axis labels don't match the corresponding figure 1 row below

Line 177. "and transmission can be attributed to the exposure measured by app". In light of known limitations, I would rephrase it as "likely attributed"

Line 183. It should be clarified that "linearly" in a log-log plot.

Line 191, figure legend: I don't see the figure that shows the maximum risk score over all exposure windows. Did you also perform an analysis where infectiousness score and proximity distance were held constant? And did you still see a strong impact by exposure duration?

Line 203. Figure 3a? there is no Figure 3b

Line 209, Figure 3 is an example where the actual number of data points/figure raw data would be informative. The confidence intervals are quite wide, suggesting a small sample size of events with high-risk scores. X-axis labels of panel A do not correspond to panels b and c. The last label in panels b and c goes to infinity. I don't think this is mathematically possible (given eq. 1 in the supplement)

Lines 271ff, please also provide actual sample sizes for these analyses. Panels e and f: what do the different bar shadings signify?

Line 317: I find extended figures 6 & 7 important, but I struggle a bit with the interpretation. I understand the baseline scenario such that any exposure duration will lead to quarantine. The point of EF6a is to show how risk-score-informed contact tracing will lead to different decisions. I am quite confused about how to read EF6a /EF7b. Why does an exposure time of 0 on the y-axis (supposedly the proximity tracing informed scenario) already lead to quarantine recommendation?

Line 349: see my general comment about "precision"

Line 356: please see my comment above re. applicability of your findings in settings with fully decentralized apps and/or a greater emphasis on privacy.

Lines 505-508: What is not quite clear to me: is P_t (probability for transmission at time t) a probability that is measured in a group (i.e. number of positive reports among all exposures notifications in a given window)? I am asking because, at the individual level, all exposure windows that are linked to a positive test will have a probability of 1, correct?

Line 519: see my comment above re. reporting of positive tests. Not reporting positive tests can have several reasons: no test was done, the test was negative, and the test was positive but not reported. Currently, the analysis seems to assume that not reporting a positive test means not being SARS-CoV-2 negative.

Supplement, eq. 2: the m^2 seems wrong. Can you please check

Referee #2 (Remarks to the Author):

This is an exceptionally clear and important analysis of the data on exposure and probability of being positive from the UK contact tracing App for COVID-19. Analyses presented are the key ones that one would like to see, and I have no major critical comments. There are a few areas of unclarity in presentation and a number of substantive points that should be expanded.

Clarity issues:

1. The x-axis in figs 1-4 varies between logarithmic, linear, and unevenly binned. This is quite confusing and, in particular with the dose-response analysis in the supplement, makes it unnecessarily hard to follow the argument and compare it to theoretical expectation. I would prefer a linear (or linearly binned) x-axis with either a logged alternative in the appendix or cutouts if needed to blow up the lower end. Some of this seems to be in the Supp figs (eg SF4) but it would be great to have a clear roadmap of which figures relate to which with different scales, and consistent scales (linear preferably) in the main text.
2. I read the paragraph at lines 467 ff. multiple times and still have a hard time understanding it. The word "individual" is ambiguous and should be replaced in all cases with "index case" or "[notified] contact" resp. for clarity. I think that would clarify nearly everything. What is unclear from this description of the data structure is: if there is no UID for the notified contact, how do the analysts know their outcome? And do they know whether the contact had a negative test, or only a positive test?
3. In EF2 I'd like to see the "low risk" section separately, as by eye it doesn't look very linear, and also confidence bounds for the slopes both for the low risk and whole thing.
4. Sup Fig 1 is not very intuitive to me. Would remove. Sup Fig 2 could be clearer with more information.
5. How is SF5b different F4abc? Other than smoothing?
6. SF6: Don't understand the negative counts. Also legend trails off in midsentence
7. Supp methods 1.2: don't understand points 2 and 6 of the notification data packet description.
8. It would be helpful to know the proportion and time-distribution of discarded notifications (where there were ≥ 2 notifications with indistinguishable personal data) and perhaps a discussion of how this might have undersampled large local gatherings and whether this affects results.
9. The argument around Supplement eq 9 is confusing. I don't really understand if there is anything more being argued than the simple point that if ascertainment is nearly independent of risk score, then the probability of testing positive should scale multiplicatively nearly as the probability of infection. This does not need 3 sentences and 3 equations. If it's more than this, please clarify.
10. I am unable to follow the argument for eq 14 in the Supplement.
11. I don't see figures whose numbers start with Y, mentioned in the Supplement
- 12.

Substantive:

1. The purpose of the heterogeneity calculations is obscure – results are not discussed in the main text, totally unclear which model if either fits the data well, and although the heterogeneity may be

a nuisance for the main purpose of the analysis, there could be some information in the analysis to understand the degree of heterogeneity in the population. Are any of the conclusions of the paper changed by the incorporation of heterogeneity?

2. Relatedly, there is not enough discussion of the ways in which the results accord with or surprise theoretical models. Roughly speaking, risk seems to be linear for low risks and saturate well below 100%. Roughly speaking, if I understand the depths of the supplement correctly, the linear part is expected and the saturation is not really discussed, though it could potentially be the ascertainment probability (not wildly off if that probability is $1/3-1/2$). This feature, and in general the conclusions one can draw from the findings about the nature of transmission, are not as well discussed as they might be. Similarly, much attention is given to maximum risk and other measures, but theoretically cumulative risk, and empirically duration, seem to be the best predictors. What is the purpose of discussing these other risk measures? Better context needs to be given to help the reader focus.

3. Also relatedly, this analysis to a first approximation finds nearly equivalent relationships between risk scores and outcomes (once background risk is subtracted off) across time and really little evidence for any heterogeneity except by urban/rural (which is shown but not explained). This seems quite strange in that the period covers the early days of vaccine introduction, the peak of 2-dose coverage, Delta, Omicron, and periods of both lockdown and school reopenings. At a minimum, vaccine coverage one would have thought would have some detectable effect in reducing the risk for a given score, by reducing infectiousness of confirmed positives and susceptibility of their contacts. But there seems to be no evidence for this. Is this a correct interpretation? If so it is perhaps the most surprising finding of the paper and deserves some discussion.

4. Supplement Lines 214ff.: Is this the methods for fig F3A? If not where is this shown? If so it would be good to show the quadratic regression. And why was binning used? It seems as if some kind of simple likelihood-based approach where the error structure is Bernoulli but the probability is linear in score would use the data better. Please try that or justify the binning approach

5. The poor performance of the machine learning approaches is probably because the relationship to observed variables is close to linear and there are few auxiliary variables for the machine learning to incorporate. How broadly generalizable do the authors think this is?

6. The specific findings about the quarantine and prophylaxis strategies are heavily dependent on non-data-based estimates of the relative cost and effectiveness of quarantine vs testing, and of other parameters for PEP. They should be more clearly qualified as illustrative in the presentation and in the abstract.

Referee #3 (Remarks to the Author):

In their manuscript "Towards precision epidemiology: digital measurement of SARS-CoV-2 transmission risk", Fraser (corresponding author) et al. (1) analyse a key performance aspect of digital contact tracing (DCT) apps, namely the correlation between intensity of exposures, as expressed by risk scores calculated by the UK DCT app, and the probability of SARS-CoV-2 transmission, as documented by app users uploading their positive test result, and (2) explore the use case of using DCT app data for estimating the transmissibility of a new virus.

This research topic is highly relevant for evaluating the performance and efficacy of a DCT app in the current pandemic and for preparedness plans re the role of such tools in future pandemics. While

DCT apps have been developed and used in many countries, most often based on the Google-Apple Exposure Notification (GAEN) framework, Fraser et al. have especially good data at their disposal, due to the high adoption and coverage of the UK DCT app and to its unique way of collecting pseudonymised user data for analysis.

The authors apply a novel methodology and present convincing results showing that the above-mentioned correlation indeed exists and that transmissibility estimation using DCT app data is possible.

Their manuscript consists of a 32-page main paper divided into a 22-page core part (with 4 figures) and a 10-page extension (with 2 tables and 8 figures), and of a 30-page supplement. In my view, their case could be still clearer and stronger, if some of the information provided in the extension and the supplement were integrated into the core part. (Examples below.)

In addition, some of the terminology should be better explained and defined. This starts with the term "contact", which is not applied for a proximity situation between individuals (as is the usual understanding), but as an abbreviation for "contact person", i.e. for the individual that had a contact (or - in this case - an exposure). This clarification (of what constitutes a "contact") is all the more important, because "contact" in "digital contact tracing", which is also frequently used in the text, does **not** imply tracing of individuals as contact persons.

Because "contacts" are a central element of their data collection and analysis, the authors need to clarify, how exactly these are constructed from filtered and bundled exposure windows. According to their manuscript, the app sends one or several "event packets" to the central server, when a user receives a notification. Each event packet contains information from one exposure window plus some information relating to the user/smartphone. As only event packets with exposure windows above the notification threshold are sent, the authors need to explain, why and how several packets can be sent for one notification. (Assumingly, an exposure window above notification threshold following an exposure window that has already triggered a notification, does not trigger a new notification, but still a packet for this subsequent exposure window is sent as part of the process triggered by the initial notification? If so, over which period of time and with which delay? And are all packets that are triggered by the same notification marked as belonging together, apart from having identical user/smartphone characteristics?)

On p 17 the manuscript states: "When only one individual with given characteristics was notified on a given day ..." How is this established? From the event packets (how?) or from other information (from the daily analytics packets)?

On p 16 the manuscript confirms about the event packet: "It contains no information about the index case to whom the contact was exposed (such information is irretrievable by the app by design) ..." However, on p 17 it says: "When more than one risky exposure window is recorded between a contact and the index case ..." and on p 4: "We grouped windows likely to have come from the same contact as a recording of the whole exposure history between that contact and the associated index case ...". How is that done? If an individual user is constructed from a set of exposure windows with unique user/smartphone characteristics, is it assumed that all exposures are to the same index case?

How does the analysis deal with the (not unlikely) possibility that an individual's exposures were to several app users that uploaded their positive test result? These exposures might be at different points in time or even simultaneously.

In this context: On p 3, the manuscript mentions "non-overlapping 'exposure windows'". To my knowledge, this is true for lasting/repetitive exposures to the same index case, but not for concurrent exposures to different index cases which may well overlap.

Because the bundling of exposure windows to construct "contacts" is so central to the analysis, the variables used for it need to be explained in full detail. In the main text, only self-reported postcode district and type of mobile device are mentioned as examples. In the supplement, 4 variables are listed (postcode district, LTLA, operating system version, device model), but it remains unclear, whether this list is comprehensive (analytics packets: "small amount of data, including the following information about the individual"; "the event packet contains the aforementioned individual-level data fields 1-4") and what the distinguishing power of these variables (in combination) is. (E.g., how many operating system versions and device models are distinguished, whether these are highly correlated, and whether the frequency of their values is unevenly distributed, with very few combinations dominating the dataset.)

NB: Re the self-reported postcode district the manuscript states: "about 20,000 individuals". Is this the average population count of postcode districts, the average count of notified app users reporting the same postcode district, or (assumably not) the total count of app users who reported a postcode district?

The same (explanation in full detail because of its central importance for the analysis) applies to the risk scoring. The authors "normalised the overall risk score such that it equals 1 for an exposure at 2 metres' distance from an index case with standard infectiousness for 15 minutes" (which is a great approach), but the details given in the Methods section do not explain what this means exactly in terms of its 3 components proximity score, duration, and infectiousness score. Even the further details in "Supplementary Methods" > "Risk scoring for the NHS COVID-19 app" leave some uncertainty. My assumption is: proximity score for 2 meters = $1/4$; standard infectiousness score = 1; thus, duration of 15 minutes = 4 (which means that the standard duration unit equalling 1 is $15/4$ minutes). If so, it would help the reader, if this is made explicit. Accordingly, the max risk score for an exposure window would be 20: proximity score for < 1 meter = 1; high infectiousness score = 2.5; max duration (30 minutes) = 8. Again, this made explicit would help the reader to better understanding the figures in the Results section.

The Extended Figure EF1 is very useful to better understand the risk scoring; consider moving it to the core part. However, it gives the false impression that the app is measuring exact distances rather than signal attenuations. It also does not explain, why the risk threshold for notification used by the app was 1.11 (10/9) rather than 1.

Some specific questions:

- On p 4, two aggregate summary metrics are introduced, namely the total duration of the exposure and the cumulative risk score, which are extensively used later on. It is stated, that "both [are] aggregated over all exposure windows of the contact." Does this really mean, that they are aggregated over all exposure windows of the contact over the entire time period from April 2021 to February 2022? If so, wouldn't it make more sense to aggregate the exposure windows of shorter time periods for the analysis? Something like this is shown in Figure F1b, but the temporal stratification apparently was done only "by the month when the notifications were received", not for the exposure windows analysed. If so, it is less surprising that "this pattern holds irrespective of season or epidemic wave". (I would indeed have expected changing probabilities of infection for pandemic phases with different dominant SARS-CoV-2 variants and an evolving immune protection through vaccination and naturally acquired immunity.)

- The same question (aggregation over which time period?) applies to the section "Empirical estimation of individuals' probability of testing positive from summary statistics" on p 16 (Methods). The previous section states: "When more than one individual with given characteristics was notified on a given day, all event packets that day with those characteristics were discarded." This would introduce a bias (exposure summaries would be systematically undercounted), if exposure windows were aggregated over time periods of several or many days; so perhaps only the exposure windows of an individual over a given day are aggregated (?) But this is not stated anywhere, as far as I saw. Also, for instance Figure F1 seems to exclude this interpretation, because it shows exposure durations far beyond 24 hours. (NB: The diagrams in row F1a and F1b are difficult to compare, because their x-axes are differently scaled. Even more so, as these are logarithmic scales, which btw is not mentioned in the label text. It seems that the max duration in F1a exceeds that in F1b, while the cumulative risk score is simply more compressed in F1a than in F1b.)

- On p 10, household contacts are defined as "exposed for at least 8 hours in a day". Does this mean, that each "contact" who had a cumulated exposure of at least 8 hours on a single day (even without any exposure on any other day) qualifies as a household contact? (This question assumes that exposure windows are aggregated over time periods longer than a day.)

Specific observations:

- On p 3: "with custom analysis of Bluetooth signal strengths exchanged between nearby devices to estimate their proximity." -> Proximity estimation is done by signal attenuation rather than strength.

- re. Supplementary Figure SF4 the label states: "the same as Figure SF3, but with observed probability of transmission on a linear rather than a logarithmic scale." However, the opposite seems to apply: SF3 = linear; SF4 = logarithmic.

Justus Benzler

Author Rebuttals to Initial Comments:

We thank the editor and referees for their careful consideration of our article. Their comments appear below formatted in black italics; our responses are in blue non-italics.

Comments by Referee #1 and our responses

Thank you for the opportunity to review the study by Ferretti and colleagues. In this article, the authors aim to demonstrate how risk scores derived from proximity tracing apps are associated with the risk of a contact person testing positive for SARS-CoV-2 within the next 14 days. This risk score integrates three components, including the proximity between two apps, the duration of proximity contact, and the likely infectiousness of the infected case (defined by the timing of contact relative to the symptom onset of the case). The authors report several noteworthy findings. First, the probability of exposure notified contacts to test positive for SARS-CoV-2 indeed increases with various risk score indicators or proximity contact duration. A more detailed analysis further identifies the length of contact as the probably most important risk component for transmission. The study further reports a disproportionately high contribution of same-household contacts (with intrinsically longer proximity contact durations). The study concludes that with further refinements of proximity tracing (e.g. by adding an "amber" warning level) and by adapting recommended measures upon exposure notification according to risk score levels, digital proximity tracing could become an important tool towards more precise (i.e. personalized) public health measures for transmission mitigation.

This is a very important study, and I applaud the authors for undertaking this detailed and thorough analysis. The study sheds further light on possible future applications of digital proximity tracing. So far, the effectiveness of such apps was mainly centered around warning exposed contacts earlier, more widely (including exposures that may be missed by conventional contact tracing), and in a more scalable fashion. The present analysis suggests that digital proximity tracing also enables more precise public health measures. Instead of recommending immediate quarantine for all exposed contacts, digital proximity tracing enables more nuanced public health measures (e.g. "test and wait") by leveraging risk scores. Therefore, the article suggests a way forward for making digital proximity tracing even more useful in future epidemics.

We thank the referee for their overall positive assessment of our article.

Data filtering and exposure window linkage/matching

=====

First and foremost, the authors search and match a large database of exposure notifications (windows) and reported positive tests. I strongly recommend that the authors present the filtering steps more concisely, provide more actual numbers (sample sizes), and discuss the limitations and potential biases more comprehensively. The sample sizes/numbers seem to be dispersed in multiple places in the main document and supplements. It would be helpful to have them all in one place, e.g. in a flow chart. Furthermore, it is unclear how the various filtering and matching procedures affect the generalizability of the main results. For example, the authors select around 53 Mio. exposure notifications (ENs). In the next step, they try to assign the ENs that belong to an "exposure window" (i.e. sequence of continuous 30-exposure windows) by a specific contact based on a probabilistic linkage. The linkage occurs based on information regarding exposure dates, iOS, mobile phone type, and zip codes provided voluntarily (in only around 20K participants). The authors claim that they have been able to uniquely assign 60% (footnote table 1) of ENs to specific exposure windows. In other words, 40% could not be assigned unambiguously, which is a substantial number. The filtering and matching steps seem quite crucial for the analysis. Hence, can the authors please provide actual numbers for matches, non-matches, and context (e.g. exposure date)? It seems to me that matching on exposure date, device, and OS alone will create large numbers of ambiguous matches - especially in high-incidence phases. Did the authors notice specific temporal patterns or associations with the incidence in "linkage success" (I am unsure whether this is what the authors modeled in subsection 1.5.2. in the supplementary materials)?...

Furthermore, the authors note a tendency for long exposure durations (which is shown in Extended Table 1 and Figures 4). What makes the authors confident that this is truly a valid finding and not an

artifact of their filtering and linkage procedures? In other words, is there supporting evidence that 1) the 40% excluded exposure notifications are not different (e.g. in terms of exposure duration or risk scores) compared with the 60% included (btw. the 60% stems from the footnote of ET1), and 2) the authors also seem to assume that the 200K persons reporting positive test results are comparable to contacts who tested positive but chose not to report their positive test result in the app.

Overall, the authors should discuss more clearly how filtering and linkage may have introduced biases and/or affected the robustness of their observations, particularly concerning their main finding of infection probability mainly depending on exposure duration.

We have:

- provided more detail in the explanation of the filtering steps in Methods (which is a more concise version of the explanation in Supplementary Materials);
- added a supplementary file of illustrative R code with toy data to show how the filtering steps work;
- expanded Extended Table 1 and added Supplementary Table S1 to show sample sizes and aspects of the data at different stages of the filtering;
- added Supplementary Figure S1 (a flowchart) and Supplementary Figure S2, and the following text in Supplementary Materials: “Grouping exposures results in a loss of about 42.8% of all packets, but the grouping process does not appear to significantly affect any of the estimates of interest in this paper. There is little difference in the average duration of exposure (3 hours and 25 minutes for all exposures versus 3 hours and 27 minutes for the grouped ones), in the fraction of exposures associated to a positive test (11.6% of all exposures versus 12.3% of the grouped ones), or in the estimate of the fraction of contacts reporting a positive test (3.5% from daily analytics data versus 3.6% from grouped exposures). Grouping exposures may introduce biases in the distribution of exposures in time and space. However, the additional noise does not seem to lead to strong distortions of the spatiotemporal distribution: the Pearson correlation between the number of all exposures versus grouped exposures by day of notification is $r=0.97$, the correlation by LTLA is $r=0.98$, the correlation by day and LTLA is $r=0.95$ (see Figure S2). Hence, these biases are unlikely to have any relevant impact on our analysis.”

To clarify two misunderstandings above:

> *zip codes provided voluntarily (in only around 20K participants)*

The self-reported postcode district is available for almost all app users; 20K is the mean population size of a postcode district. We have clarified the text here.

> *matching on exposure date, device, and OS alone will create large numbers of ambiguous matches*

In the Methods section, for brevity we named only two examples of individual-level fields used for matching—postcode district and OS—in addition to exposure date. In the Supplementary Material we named all four fields: postcode district, OS, device model and lower-tier local authority. For clarity we now mention all four data fields in Methods too. Grouping all app data by these fields combined does not typically result in groups of only a single individual; however, grouping only those individuals who were notified on the same date by these fields results, 60% of the time, in only one individual with a particular combination of these fields. That is the subset of roughly 7 million individuals considered in this study.

Along similar lines, the authors report that around 200K (out of 7 Mio) exposure contacts reported follow-up positive test results within the likely exposure window. Again, the selection and matching of exposure windows matter here because these processes influence probability calculations for testing

positive after exposure. My questions here: What was the denominator for the probability of infection analysis? Was it the 7 Mio contacts - assuming that those not reporting a positive result never tested positive during the exposure window? A positive test may be missing for multiple reasons: no test was done, the test was negative, the test was positive but not reported, or the test was reported but the information was not transmitted due to technical errors. Several assumptions surrounding the accuracy of positive tests are reported in the supplementary materials (e.g. lines 104-131 or section 1.7). But they do not appear to investigate (or I fail to see it) whether the underlying assumption that the absence of positive tests means test negativity (in a mathematical sense) is justified.

The referee is correct about our assumptions and why they result in underestimates; this was explained in the Results section:

“The data also indicated whether the contact reported a positive SARS-CoV-2 test through the app during an interval beginning with their notification and ending 14 days after the exposure. The fraction of contacts doing so defines the observed probability of infection. This outcome is a proxy for the true probability of being infected, though it is smaller: an unknown but likely appreciable fraction of infected app users either do not seek a test, or do not report their positive result through the app, or report it outside of the aforementioned interval. As a reference, the number of infections in adults in the same period in the UK was estimated to be 2-3 times greater than the number of confirmed cases [22].”

Also later in Results:

“Both of these observed probabilities [infection and transmission] are lower than the corresponding true probabilities due to unreported infections.”

For extra clarity we now repeat the point in Discussion:

“Also, testing was not compulsory for contacts, therefore infections were likely under-reported and absolute transmission rates must be interpreted with caution. Biases in testing or reporting, such as an increased propensity to get tested after a close contact tested positive, may also affect our results.”

We have also changed terminology from ‘observed probability of infection/transmission’ to ‘probability of reported infection/transmission’.

Technical differences between the NHS Covid-19 app and fully decentralized GAEN apps

=====
This may be a bit picky, but in the Introduction of the Supplementary Materials the UK NHS Covid-19 app is portrayed as relying on the Google Apple Exposure Notification (GAEN) framework. While this may technically be true (I am unable to say so), the UK app deviates from other GAEN apps in important ways. Specifically, the UK app stores exposure notifications in a central server, which enables the matching of exposure notifications with positive tests. Most apps I am familiar with have an entirely decentralized architecture, where exposure notifications are entirely analyzed on mobile phones, and no data is sent back to centralized servers.

The architectural differences are relevant for the discussion of future directions of leveraging digital proximity tracing for more precise public health actions. The final sentence and recommendation (line 429, as well as line 356ff.) to integrate risk scores into public health toolboxes may be quite controversial in some settings. I remember quite heated debates with international colleagues about whether and how digital proximity tracing apps should also be turned into data collection tools to learn more about transmission dynamics (as proposed in line 401). I think it may be important to acknowledge that some jurisdictions may not want to collect exposure notifications in a centralized system. Thus, how would the author's findings still be useful to make contact tracing more precise in systems and/or in settings with fully decentralized digital proximity tracing? The only way I could see this work if risk scores and alert levels are only stored locally on mobile phones, is if contact tracers ask app users to send a screenshot of alert levels (amber or red) or the like.

We confirm that the NHS COVID-19 app is based on the GAEN framework. The architecture of the app is fully decentralised, and only anonymous analytics data packets are sent back to central servers. In the GAEN framework there is no limitation on sending anonymous data back to a central server for the purpose of monitoring the correct technical behaviour and the effectiveness of contact tracing apps; however, it is true that this has been implemented in very few cases during the pandemic. App analytics for evaluating its proper functioning and effectiveness has been explicitly asked by the Information Commissioner's Office in the UK, and the need for such evaluations has been recently mandated by the Council of Europe.

(In principle, some evaluations may also be performed through decentralised approaches that do not share individual data with a central server, in the style of Baker et al

<https://www.pnas.org/doi/10.1073/pnas.2106548118>. The feasibility and implementation of these options is beyond the scope of this paper.)

Overemphasis of the word "precision" in some contexts (in my opinion)

=====

*I very much agree with the aim of precision public health, and the author's work is a really important milestone to make contact tracing *more precise*. However, I have some issues with the claim that using risk scores equates to precision contact tracing (line 55, line 401 which speaks about high-precision risk assessment) or line 35 in the summary. But it is still a very long way to real precision - as the authors acknowledge themselves in lines 403ff...*

Lines 54 and 133: Examples where I think the "precision" aspect should be toned down a bit, e.g. "approach towards precision epidemiology" in line 54...

Line 349: see my general comment about "precision"

We agree that using risk scores does not equate to precision contact tracing, and did not intentionally give this impression in the text. We have removed all mention of precision epidemiology or contact tracing throughout the manuscript, except in the new closing paragraph of discussion, where we explain more clearly what we mean by this term and why this work is progress towards (but not achievement of) this goal.

The role of aerosol transmission

=====

The initially proposed heuristic of 2m /15min for contact tracing, as well as distancing rules for protection, were based on the assumption that SARS-CoV-2 transmission would mainly occur via droplets. Meanwhile, there is also substantial evidence for aerosol transmission of SARS-CoV-2 (e.g. <https://www.bmj.com/content/377/bmj-2021-068743>,

[https://www.thelancet.com/journals/lancet/article/PIIS0140-6736\(21\)00869-2/fulltext?ref=vc.ru](https://www.thelancet.com/journals/lancet/article/PIIS0140-6736(21)00869-2/fulltext?ref=vc.ru)).

Against this backdrop, can the authors speculate whether and how aerosol transmission (as opposed to "proximity-based" risks) may have influenced their findings and their generalizability? It seems to me that aerosol transmission would not be captured by this analysis unless some relevant proximity (leading to a risk score above the threshold) was also involved. For instance, could the author's finding of substantial infection risks for "longer exposures at greater distances" (line 28f, line 365) reflect aerosol transmission of SARS-CoV-2? On another note, the inability of digital proximity tracing apps to accurately measure aerosol transmission is yet another reason why I would refrain from calling risk score-augmented manual contact tracing "high precision contact tracing".

The risk score for the app was developed quite early in the pandemic, even before the GAEN protocol was proposed, when there was limited reliable information on the mode of transmission. For this reason, the proximity score has a quite large tail (decreasing as the inverse of the square distance) which is not dissimilar to the one expected under aerosol transmission. In fact, Extended Figure 1 shows how longer exposures at larger distances (~4 metres) would still be traced by the app, at least in principle.

At the time the score was developed, we supposed that the risk assessment would have been continuously improved with increased quantitative knowledge and measurements of viral transmission, and that the app risk score computation would have been updated accordingly. However, this never happened. Hence, while the risk threshold was updated several times during the pandemic (but always close to the estimated risk at about 15 minutes at 2 metres), the initial risk score function (already appropriate for aerosol transmission) was never changed.

While our results (especially Figure 3) show that our proximity/risk score reflects the actual risk of transmission, and therefore seem to suggest that SARS-CoV-2 transmission has a substantial aerosol component, the evidence is indirect and we prefer not to speculate on the subject here. We hope to be able to reassess this topic in more detail in the future.

Minor comments

Line 47: Can you add references for the "social and economic costs" of conventional epidemic control measures?

Done

Line 57: digitized the process of contact tracing. I am sure persons involved in manual contact tracing would object to this statement. Manual contact tracing assesses more information than "just" proximity.

We do not understand the referee's argument here, unless two separate points are being made: whether or not manual contact tracing assesses more information than proximity is unrelated to whether the process is digital.

To clarify, we did not state that manual contact tracing just considers proximity at line 57 or elsewhere. Indeed we later state that contact tracing generally was based on both proximity and duration, and that contextual information about exposure—missing from app-based tracing—may be used to assess risk for manual contact tracing.

Regarding the first point, apps digitise the processes of measuring risk, storing it in memory, recalling it, comparing it to the thresholds, and using it to communicate from index to their contacts. Computers may be involved in manual contact tracing, for example to type and save contact details in a database, but none of the aforementioned steps are done 'digitally'.

Figure F1: please use consistent labels, i.e. risk scores rather than risk predictor...

We have updated the label to simply 'score'. (Throughout our article we use the term 'risk score' to refer specifically to the quantity calculated by the app for a single exposure window. In this plot we show three different quantities: the maximum risk score over all windows an individual has, the cumulative risk score, and the cumulative duration. These three quantities can be considered as different types of score or risk predictor, none of which is the 'risk score' referred to elsewhere.)

...The second panel from left, top row: values of x-axis labels don't match the corresponding figure 1 row below...

Corrected as suggested.

...Line 177. "and transmission can be attributed to the exposure measured by app". In light of known limitations, I would rephrase it as "likely attributed"

We agree with the reviewer that the process of attribution is always subject to limitations. For clarity we now use the verb attribute in an active rather than a passive voice, i.e. 'we attributed' rather than 'can be attributed', and we have replaced 'transmission caused by the recorded exposure' by 'transmission we attributed to the recorded exposure'.

Line 183. It should be clarified that "linearly" in a log-log plot.

Here we use "linearly" for the first part of the plot (orange regression). The trend there is linear on a linear scale, as well as on a log-log scale, i.e. the regression coefficient is indistinguishable from 1. We added the coefficients to the regressions in the log-log plot to clarify this explicitly.

Line 191, figure legend: I don't see the figure that shows the maximum risk score over all exposure windows. Did you also perform an analysis where infectiousness score and proximity distance were held constant? And did you still see a strong impact by exposure duration?

Apologies, we removed this alternative figure as it contributed little extra. This analysis shows how duration has a strong impact as mean risk score is held constant; we did not analyse the effect of the infectiousness score because most risky exposures were to an index case classified as the higher of the two infectiousness levels (thanks to the relatively short delays of the testing system in England and Wales), giving little variability to explore.

Line 203. Figure 3a? there is no Figure 3b

At this point in the text we previously referred to Figure F3a, which did previously exist (as did Figure 3b). For space we have now removed panels b and c of Figure 3 to an Extended Figure, and now refer simply to Figure 3.

Line 209, Figure 3 is an example where the actual number of data points/figure raw data would be informative. The confidence intervals are quite wide, suggesting a small sample size of events with high-risk scores. X-axis labels of panel A do not correspond to panels b and c. The last label in panels b and c goes to infinity. I don't think this is mathematically possible (given eq. 1 in the supplement)

We now provide Supplementary files with the raw data or data points for all figures in the main text. The referee is right that the maximum value observed is finite; we used infinity as the upper bound for the largest bin to indicate that the bin is not bounded from above (alternatively, all upper bounds above the largest observed value are equivalent). We no longer report an upper bound here, as part of harmonising the x axes of these plots (previously Figure 3 panels b and c, now Extended Figure EF8) per Referee 2's first clarity point.

Lines 271ff, please also provide actual sample sizes for these analyses. Panels e and f: what do the different bar shadings signify?

Sample sizes (i.e. number of contacts and transmissions) are now reported in Extended Table 1. Since they are the same for all analyses (i.e. 52,979,100 exposure windows, 7,047,541 contacts, 239,683 contacts reporting a positive test after notification), we decided not to report them everywhere. We added the sentence "Extended Table E1 summarises sample sizes for the final dataset analysed in this paper."

Sample sizes for individual bins, as well as other information about the plots, have now been added as Supplementary Data files.

We now explain that the different bar shadings signify the confidence intervals associated with the calculation of the background risk.

Line 317: I find extended figures 6 & 7 important, but I struggle a bit with the interpretation. I understand the baseline scenario such that any exposure duration will lead to quarantine. The point of EF6a is to show how risk-score-informed contact tracing will lead to different decisions. I am quite confused about how to read EF6a /EF7b. Why does an exposure time of 0 on the y-axis (supposedly the proximity tracing informed scenario) already lead to quarantine recommendation?

We changed the format of these figures to make them easier to understand. The baseline scenario is now to quarantine contacts that were exposed for a duration greater than a given threshold. For different values of this threshold, we show less costly strategies with the same effectiveness (E6) or more effective strategies with the cost (E7).

Line 356: please see my comment above re. applicability of your findings in settings with fully decentralized apps and/or a greater emphasis on privacy.

Our comment here on the ease of knowledge transfer refers specifically to the epidemiological finding of duration being a nuanced and important predictor of risk (i.e. beyond simply ‘less or more than 15 minutes’). This finding is useful beyond fully decentralised apps; indeed our point is that it is useful without any kind of app, for example in manual contact tracing.

Lines 505-508: What is not quite clear to me: is P_t (probability for transmission at time t) a probability that is measured in a group (i.e. number of positive reports among all exposures notifications in a given window)? I am asking because, at the individual level, all exposure windows that are linked to a positive test will have a probability of 1, correct?

Apologies for the confusion: P_t is not the probability of transmission at time t . At this point in the text we previously explained:

“ P_t (...) is the probability of transmission during the given recorded window (followed by reporting a positive test)”

we have clarified this to

“ P_i (i 's n th window) is the probability of transmission during i 's n th window (followed by reporting a positive test)”

This section describes a statistical model, of which P_t for a given type of window is a population-level risk parameter to be estimated (not empirically determined for given individuals). We have included three extra sentences to this section to clarify.

Line 519: see my comment above re. reporting of positive tests. Not reporting positive tests can have several reasons: no test was done, the test was negative, and the test was positive but not reported. Currently, the analysis seems to assume that not reporting a positive test means not being SARS-CoV-2 negative.

We added caveats in Results and Discussion, see our response above.

Supplement, eq. 2: the m^2 seems wrong. Can you please check

It is correct: ρ (rho) has dimension [length], and so to obtain a dimensionless score $s(\rho)$ that is proportional to ρ^2 , the proportionality constant must have dimension [length]².

Comments by Referee #2 and our responses

This is an exceptionally clear and important analysis of the data on exposure and probability of being positive from the UK contact tracing App for COVID-19. Analyses presented are the key ones that one would like to see, and I have no major critical comments. There are a few areas of unclarity in presentation and a number of substantive points that should be expanded.

We thank the referee for their overall positive assessment of our article.

Clarity issues:

1. The x-axis in figs 1-4 varies between logarithmic, linear, and unevenly binned. This is quite confusing and, in particular with the dose-response analysis in the supplement, makes it unnecessarily hard to follow the argument and compare it to theoretical expectation. I would prefer a linear (or linearly binned) x-axis with either a logged alternative in the appendix or cutouts if needed to blow up the lower end. Some of this seems to be in the Supp figs (eg SF4) but it would be great to have a clear roadmap of which figures relate to which with different scales, and consistent scales (linear preferably) in the main text.

Given the wide range of variability of duration and cumulative risk score, we moved all x axes to logarithmic scale. The only exception is Figure 3, where the linear trend is clearer on a linear scale.

2. I read the paragraph at lines 467 ff. multiple times and still have a hard time understanding it. The word “individual” is ambiguous and should be replaced in all cases with “index case” or “[notified] contact” resp. for clarity. I think that would clarify nearly everything. What is unclear from this description of the data structure is: if there is no UID for the notified contact, how do the analysts know their outcome? And do they know whether the contact had a negative test, or only a positive test?

We replaced all references to ‘the individual’ here with ‘the contact’ as suggested.

The way in which we are able to determine the test-positive outcome for contacts after they are notified was explained briefly in Methods:

“If an individual reports a positive test in the app during the ‘observation interval’—starting with their notification and ending 14 days after the exposure—the same event packets are sent once more to the central server, flagged as associated with a positive test. This flag allows the linkage between exposure windows and the binary outcome of positive test reported or not.”

And in greater detail in the Supplementary Methods Section 1.2. We have now appended the following to the brief Methods explanation: “because we can see which event packets were sent a second time with all data identical except for this flag, and which events were not”. (The validity of this assumption was tested and discussed in Supplementary Methods section 1.3.)

We do not analyse whether contacts report negative tests (as far as we know, data on negative tests is underascertained to extreme levels, since there was no push to report negative tests). Our outcome is simply whether they report a positive test through the app during the observation interval.

3. In EF2 I'd like to see the “low risk” section separately, as by eye it doesn't look very linear, and also confidence bounds for the slopes both for the low risk and whole thing.

We now report the regression coefficients and the confidence intervals for the slope of the log-log plot, which overlap 1 with decent precision, i.e. indicating a trend compatible with linearity.

4. *Sup Fig 1 is not very intuitive to me. Would remove. Sup Fig 2 could be clearer with more information.*

We have removed both of these figures, as they provided only highly simplified illustrations of the app data process without adding any clarity beyond what was discussed in words.

5. *How is SF5b different F4abc? Other than smoothing?*

As well as using binning rather than smoothing, this supplementary figure also shows the distributions for all reported infections (i.e. before subtraction of the background), which the main text figure does not show for simplicity.

6. *SF6: Don't understand the negative counts. Also legend trails off in midsentence*

Negative counts illustrate how the naive “full correction” is an overcorrection, and why we use the maximum-likelihood correction instead, as now clarified in the caption.

7. *Supp methods 1.2: don't understand points 2 and 6 of the notification data packet description.*

We clarified both points.

8. *It would be helpful to know the proportion and time-distribution of discarded notifications (where there were ≥ 2 notifications with indistinguishable personal data) and perhaps a discussion of how this might have undersampled large local gatherings and whether this affects results.*

Please see our response to the first (essentially identical) suggestion made by Referee 1. We agree that it is an important point. We added new results and discussion in Supplementary Methods Section 1.3, showing that the grouping process does not appear to lead to significant epidemiological biases, although some biases are clearly present for combinations of common phone models and OS versions, large postcode districts and high COVID-19 prevalence.

We agree that there may be some undersampling of large local gatherings, though it is likely to be a minor effect for the same reasoning discussed in the new Supplementary Methods section about simultaneous notifications from multiple index cases.

9. *The argument around Supplement eq 9 is confusing. I don't really understand if there is anything more being argued than the simple point that if ascertainment is nearly independent of risk score, then the probability of testing positive should scale multiplicatively nearly as the probability of infection. This does not need 3 sentences and 3 equations. If it's more than this, please clarify.*

We have removed this argument, which was not helpfully contributing.

10. *I am unable to follow the argument for eq 14 in the Supplement.*

Here we explained that heterogeneities between contacts (indexed by i) in their susceptibility h_i and ascertainment a_i may be correlated with each other and with the individual's cumulative risk r_i^{cum} , but that we approximate these correlations as zero for tractability, since h_i and a_i are unobserved. We have now added to this explanation that this approximation implies the joint probability for h_i and a_i given r_i^{cum} is $p(h_i, a_i | r_i^{\text{cum}}) = p(h_i) p(a_i)$, and using this to marginalise over h_i and a_i in equation 13 gives equation 14 (now renumbered to 16 with the two implicit steps in between now made explicit).

11. *I don't see figures whose numbers start with Y, mentioned in the Supplement*

Apologies, we have corrected the numbering in line 303.

Substantive:

1. The purpose of the heterogeneity calculations is obscure – results are not discussed in the main text, totally unclear which model if either fits the data well, and although the heterogeneity may be a nuisance for the main purpose of the analysis, there could be some information in the analysis to understand the degree of heterogeneity in the population. Are any of the conclusions of the paper changed by the incorporation of heterogeneity?

We included the heterogeneities as a robustness/sensitivity analysis. We clarified this in the text: “The relationship is robust with respect to individual heterogeneities or underreporting of positive tests among contacts (Extended Figure E3)” and “As a robustness check, we developed likelihoods based on frailty models with several sources of heterogeneity among index-contact pairs in the model”. We show the corresponding results for the Maximum Likelihood estimates of risk as a function of (binned) risk score in Extended Figure E3, and we also added Supplementary Table S5 with AIC values to show which models fit the data better, as well as ML parameter values for these models. It is tempting to interpret the heterogeneities in the population in terms of actual heterogeneities in risk. However, we do not feel confident enough about the (multiple) sources of these heterogeneities. We think there is a risk of overinterpreting the results, and we prefer to refrain from doing so.

2. Relatedly, there is not enough discussion of the ways in which the results accord with or surprise theoretical models. Roughly speaking, risk seems to be linear for low risks and saturate well below 100%. Roughly speaking, if I understand the depths of the supplement correctly, the linear part is expected and the saturation is not really discussed, though it could potentially be the ascertainment probability (not wildly off if that probability is 1/3-1/2). This feature, and in general the conclusions one can draw from the findings about the nature of transmission, are not as well discussed as they might be. Similarly, much attention is given to maximum risk and other measures, but theoretically cumulative risk, and empirically duration, seem to be the best predictors. What is the purpose of discussing these other risk measures? Better context needs to be given to help the reader focus.

We comment on the linear dependence for short durations, which is expected in theory but never fully appreciated nor shown empirically before as far as we are aware. Concerning saturation, since (i) we do not reach full saturation and (ii) there is ascertainment bias, we do not feel confident speculating on it. As pointed out by the reviewer, due to ascertainment bias, we may well be simply seeing the classical saturation $probability_of_transmission \sim 1 - exp(-cumulative_risk)$. However, similar but more complex versions are possible with transmission rate heterogeneity: $probability_of_transmission \sim 1 - Laplace_transform_of_heterogeneities(cumulative_risk)$. We added a comment about this possibility in Supplementary Methods Section 1.5.

We discuss maximum risk because it is what the app actually uses: notification occurs if and only if any window has risk over the threshold, i.e. if $\max(risk)$ is over the threshold. Cumulative risk is the better predictor theoretically and, as these results show, empirically. Duration is also better than maximum risk, and easier to translate to assessment of risk without apps. Hence all three merit discussion. We now emphasise this where the three metrics are introduced to better provide context for what follows.

3. Also relatedly, this analysis to a first approximation finds nearly equivalent relationships between risk scores and outcomes (once background risk is subtracted off) across time and really little evidence for any heterogeneity except by urban/rural (which is shown but not explained). This seems quite strange in that the period covers the early days of vaccine introduction, the peak of 2-dose coverage, Delta, Omicron, and periods of both lockdown and school reopenings. At a minimum,

vaccine coverage one would have thought would have some detectable effect in reducing the risk for a given score, by reducing infectiousness of confirmed positives and susceptibility of their contacts. But there seems to be no evidence for this. Is this a correct interpretation? If so it is perhaps the most surprising finding of the paper and deserves some discussion.

For this study our data starts from April 2021. Indeed, vaccines did have a strong impact which was detectable through the NHS app, but this occurred before the period analysed here. It is visible as a large decrease in January-April 2021 in the fraction of notified contacts that reported a positive test (Figure 5c in <https://doi.org/10.1038/s41467-023-36495-z>, which uses daily analytics from the NHS app rather than exposure data). The impact of initial vaccination is therefore already included in the probabilities of transmission presented here. School closures/openings changed little during the period of study: "All primary schools in England reopened on 8 March whilst secondary schools opened in a staggered manner beginning on that date [Wikipedia]." Also, only individuals aged 16 and over were eligible to use the app; such individuals could be infected by under-16s of course, but these exposures would not be recorded by the app, and so they contribute to the background risk which we subtract. Theoretically, we would not expect our background-correct results to vary much with lockdowns (or indeed school closures) if our background correction worked as intended: these results are for how variability in the nature of an exposure to a single individual predicts the outcome of transmission (through the proxy of reported positive tests). Lockdowns should decrease the number of low-risk exposures that individuals have on average (and possibly increase the number of high-risk exposures if individuals mix for longer times with their household), but not decrease the probability of transmission given a particular kind of exposure.

Some changes in viral variants could be detected with app data. For example in December 2021 we submitted an internal report to UKHSA with an estimate of the relative secondary attack rate of Omicron: through a regression analysis of app data, we estimated that the SAR for Omicron (BA.1) was three times higher than the SAR for Delta, in agreement with estimates from manual contact tracing from the same period. The agreement between manual contact tracing and app data supports the reliability of our data to measure relative values of SAR.

However, after an initial period in December, the replacement of Delta by Omicron does not seem to have led to an increased SAR. This is consistent with known temporal variation in the reproduction number R_t in the UK, but a proper illustration and discussion of this issue would take far more space than we can present in this manuscript. We can only speculate that this surprising behaviour may possibly be caused by a combination of behavioural changes and rapid reduction in the number of susceptible individuals, but we do not have any further insight on the topic from app data.

Regarding rural/urban differences, we have moved the result to Extended Figure 8. While it is easier to speculate that the weekend effect in Figure E8a could be explained in terms of riskier settings (e.g. meals, pubs/restaurants/venues), which have been well documented for COVID-19, we do not have any simple explanation for the lower risk in conurbations in Figure E8b. It may be due to different interactions, e.g. the same measurements could correspond to occasional close proximity in conurbations versus direct social interactions (which tend to be riskier) in rural areas. But we have no data or evidence to support this.

4. Supplement Lines 214ff.: Is this the methods for fig F3A? If not where is this shown? If so it would be good to show the quadratic regression. And why was binning used? It seems as if some kind of

simple likelihood-based approach where the error structure is Bernoulli but the probability is linear in score would use the data better. Please try that or justify the binning approach

The paragraph from Supplement lines 214 was indeed confusing; we have rewritten it. The quadratic regression was simply used to provide a reference value for the risk at 2 meters for 15 minutes, i.e. the grey dots in Figures 1 and 3.

The Methods subsection ‘Statistical modelling of the per-exposure-window probability of transmission’ explains that it is for Figure F3A, now named Figure 3. This method is a likelihood-based Bernoulli model. In the likelihood we did not assume that the transmission probability is linear in risk score—instead allowing the data to reveal this linearity—because this serves to validate this risk score being epidemiologically meaningful. We previously alluded to this in Discussion and have now said so more explicitly in Results and Methods. After seeing that the data support linearity, indeed we can say that a model based on the assumption of linearity would provide a slightly more precise estimate of the slope (as a check, we run such an approximate model reducing it to a log-binomial regression, finding basically identical results). However, the large size of this dataset allows precise estimation even with flexible models like the one used.

5. The poor performance of the machine learning approaches is probably because the relationship to observed variables is close to linear and there are few auxiliary variables for the machine learning to incorporate. How broadly generalizable do the authors think this is?

We agree that it is probably due to the lack of auxiliary variables related e.g. to the setting of the exposures. Given this limitation, we removed the related comments from the Discussion as part of shortening of the manuscript.

6. The specific findings about the quarantine and prophylaxis strategies are heavily dependent on non-data-based estimates of the relative cost and effectiveness of quarantine vs testing, and of other parameters for PEP. They should be more clearly qualified as illustrative in the presentation and in the abstract.

The referee is correct; the corresponding figures were not included in the main text because they are only shown for illustrative purposes. We have de-emphasised the results further by moving the prophylaxis figure to Supplementary Discussion and removing any detailed mention from the main text. We also changed the text of the other example (amber notifications, in main text and Extended Figures) to reflect the fact that the results are simply illustrative.

Comments by Referee #3 and our responses

In their manuscript "Towards precision epidemiology: digital measurement of SARS-CoV-2 transmission risk", Fraser (corresponding author) et al. (1) analyse a key performance aspect of digital contact tracing (DCT) apps, namely the correlation between intensity of exposures, as expressed by risk scores calculated by the UK DCT app, and the probability of SARS-CoV-2 transmission, as documented by app users uploading their positive test result, and (2) explore the use case of using DCT app data for estimating the transmissibility of a new virus.

This research topic is highly relevant for evaluating the performance and efficacy of a DCT app in the current pandemic and for preparedness plans re the role of such tools in future pandemics. While DCT apps have been developed and used in many countries, most often based on the Google-Apple Exposure Notification (GAEN) framework, Fraser et al. have especially good data at their disposal, due to the high adoption and coverage of the UK DCT app and to its unique way of collecting pseudonymised user data for analysis.

The authors apply a novel methodology and present convincing results showing that the above-mentioned correlation indeed exists and that transmissibility estimation using DCT app data is possible.

We thank the referee for their overall positive assessment of our article.

Their manuscript consists of a 32-page main paper divided into a 22-page core part (with 4 figures) and a 10-page extension (with 2 tables and 8 figures), and of a 30-page supplement. In my view, their case could be still clearer and stronger, if some of the information provided in the extension and the supplement were integrated into the core part. (Examples below.)

We respond to the suggestions individually below. Our main text previously being 800 words over the maximum limited our ability to incorporate more content there.

*In addition, some of the terminology should be better explained and defined. This starts with the term "contact", which is not applied for a proximity situation between individuals (as is the usual understanding), but as an abbreviation for "contact person", i.e. for the individual that had a contact (or - in this case - an exposure). This clarification (of what constitutes a "contact") is all the more important, because "contact" in "digital contact tracing", which is also frequently used in the text, does **not** imply tracing of individuals as contact persons.*

The meaning of “contact” is subtle. In the term “contact tracing”, digital or manual, both common meanings of “a contact”—an exposure event or an exposed individual—make sense. For the former, we are talking about a tracing process that is based on such events; for the latter, we are talking about a process that traces such individuals. “contact tracing” and “tracing contacts” are thus synonymous when “contact” means an exposed individual. This terminology makes sense in the same way that the term “partner notification” does in the field of HIV: the person then the process. In this manuscript we used the standard terminology for SARS-CoV-2 in the UK, where “contact” refers to the individual rather than the event (see e.g. <https://www.gov.uk/guidance/nhs-test-and-trace-how-it-works>). For clarity we now define our meaning of “contact” in the abstract.

Because "contacts" are a central element of their data collection and analysis, the authors need to clarify, how exactly these are constructed from filtered and bundled exposure windows...

On p 16 the manuscript confirms about the event packet: "It contains no information about the index case to whom the contact was exposed (such information is irretrievable by the app by design) ..." However, on p 17 it says: "When more than one risky exposure window is recorded between a contact and the index case ..." and on p 4: "We grouped windows likely to have come from the same contact as a recording of the whole exposure history between that contact and the associated index case ...". How is that done?...

On p 17 the manuscript states: "When only one individual with given characteristics was notified on a given day ..." How is this established? From the event packets (how?) or from other information (from the daily analytics packets)?

Please see our response to the first suggestion made by Referee 1. Grouping is done on the basis of some fields that should be the same for all packets from a given exposure (postcode district, phone model, OS version, local authority, date of notification). We establish that only one individual was notified that day based on daily analytics packets. We added more details in both Methods and Supplementary, including flow diagrams and summary tables, as well as a proper analysis of the accuracy of the grouping process.

...According to their manuscript, the app sends one or several "event packets" to the central server, when a user receives a notification. Each event packet contains information from one exposure window plus some information relating to the user/smartphone. As only event packets with exposure windows above the notification threshold are sent, the authors need to explain, why and how several packets can be sent for one notification. (Assumingly, an exposure window above notification threshold following an exposure window that has already triggered a notification, does not trigger a new notification, but still a packet for this subsequent exposure window is sent as part of the process triggered by the initial notification? If so, over which period of time and with which delay? And are all packets that are triggered by the same notification marked as belonging together, apart from having identical user/smartphone characteristics?)

Consider a single exposure event between a given index case and contact. This event is analysed, at the time the index case reports their positive test, separately in multiple 30-minute exposure windows. If at least one of these is over the threshold for notification the contact is notified. One event packet is sent for each window over the threshold (a 'risky' window); in general, there are multiple risky windows from a single exposure event and so multiple event packets are sent simultaneously. We have restructured the first few paragraphs of Results to make this clearer.

...If an individual user is constructed from a set of exposure windows with unique user/smartphone characteristics, is it assumed that all exposures are to the same index case? How does the analysis deal with the (not unlikely) possibility that an individual's exposures were to several app users that uploaded their positive test result? These exposures might be at different points in time or even simultaneously.

The referee is correct that we didn't explain why the great majority of notifications were likely caused by exposure to a single infected case. We now present a simplified but robust model of the infection/notification process in Supplementary Methods Section 1.3.2, showing how exposures to multiple individuals (most likely caused by superspreading events occurring among acquaintances of the contact) may contribute to the same notification in about 1% of cases.

In this context: On p 3, the manuscript mentions "non-overlapping 'exposure windows'". To my knowledge, this is true for lasting/repetitive exposures to the same index case, but not for concurrent exposures to different index cases which may well overlap.

This is correct. In fact, that sentence refers explicitly to an exposure to a specific index case. Multiple exposures to different index cases are likely to represent a very small part of the dataset, as explained in the previous answer.

Because the bundling of exposure windows to construct "contacts" is so central to the analysis, the variables used for it need to be explained in full detail. In the main text, only self-reported postcode district and type of mobile device are mentioned as examples. In the supplement, 4 variables are listed (postcode district, LTLA, operating system version, device model), but it remains unclear, whether this list is comprehensive (analytics packets: "small amount of data, including the following information about the individual"; "the event packet contains the aforementioned individual-level data fields 1-4") and what the distinguishing power of these variables (in combination) is. (E.g., how many operating system versions and device models are distinguished, whether these are highly correlated, and whether the frequency of their values is unevenly distributed, with very few combinations dominating the dataset.)

These four variables are comprehensive for defining our 'individual-level characteristics'; for clarity we now list all four of these variables in Methods as well as Supplementary, and in Supplementary we have added:

“Henceforth we refer to the four variables above as the ‘individual-level’ characteristics or data, because they are expected to remain constant for a given individual over the time scale of one round of exposure, contact tracing and testing. For other data fields this is not true, for example those recording technical functioning of the app, and the data field indicating whether or not the user was notified that day of a risky exposure.”

Note that as explained elsewhere, one contact is not defined simply by grouping the full dataset by these four variables: we require the analytics packets sent daily by each user to contain only a single individual notified of risky exposure on a given date with a unique combination of these four variables.

There is an uneven distribution of these individual-level characteristics, for example some device models are much more common than others. There are some strong correlations within the individual-level characteristics, particularly between the device model and operating system (for example iOS only on iPhones); in practise this simply reduces their distinguishing power relative to a counterfactual in which there is no correlation (e.g. any operating system can be chosen randomly on every device). The measure of distinguishing power that is relevant for this analysis is that 60% of event packets could be grouped as from a single putative individual (exactly one app user with those individual-level characteristics was notified on that date).

NB: Re the self-reported postcode district the manuscript states: "about 20,000 individuals". Is this the average population count of postcode districts, the average count of notified app users reporting the same postcode district, or (assumingly not) the total count of app users who reported a postcode district?

We have reworded for clarity: this is the mean population size of a postcode district (we report this to give an idea of the level of geographical granularity).

The same (explanation in full detail because of its central importance for the analysis) applies to the risk scoring. The authors "normalised the overall risk score such that it equals 1 for an exposure at 2 metres' distance from an index case with standard infectiousness for 15 minutes" (which is a great approach), but the details given in the Methods section do not explain what this means exactly in terms of its 3 components proximity score, duration, and infectiousness score. Even the further details in "Supplementary Methods" > "Risk scoring for the NHS COVID-19 app" leave some uncertainty. My assumption is: proximity score for 2 meters = 1/4; standard infectiousness score = 1; thus, duration of 15 minutes = 4 (which means that the standard duration unit equalling 1 is 15/4 minutes). If so, it would help the reader, if this is made explicit. Accordingly, the max risk score for an exposure window would be 20: proximity score for < 1 meter = 1; high infectiousness score = 2.5; max duration (30 minutes) = 8. Again, this made explicit would help the reader to better understanding the figures in the Results section.

Indeed the maximum possible risk score for a window is 20; we now note this at first introduction of the risk scores to give readers a sense of scale as suggested.

Defining standard units for all three of the factors contributing multiplicatively to the overall score is arbitrary, because one of the factors could be multiplied by X and another divided by X, for any X, with no effect. All that matters is the standard unit for the overall product of the factors - our normalisation referred to above. For clarity, we have added the following just before Supplementary equation 1:

“The overall scaling of the risk score for a window (and thus of each of its three multiplicative components) is arbitrary: action is only taken depending on its size relative to the threshold for notification, which can be scaled correspondingly. For ease of interpretation, we normalised all risk

scores by the risk score for a 15-minute encounter with an infected individual of standard infectiousness at 2 metres' distance.”

Supplementary equation 1 then specifies how the risk score is calculated with this normalisation, for any combination of proximity, duration and infectiousness values the reader might be interested in. We note the three special cases of the reference value of 1 (15-minutes exposed to a case with standard infectiousness at 2 metres' distance), the notification threshold of 1.11, and the maximum of 20 (30 minutes exposed to a case with high infectiousness at less than 1 metres' distance).

The Extended Figure EF1 is very useful to better understand the risk scoring; consider moving it to the core part. However, it gives the false impression that the app is measuring exact distances rather than signal attenuations. It also does not explain, why the risk threshold for notification used by the app was 1.11 (10/9) rather than 1.

We have added to EF1 “These boundaries apply in theory, though in practice distances are estimated with imperfect precision.”

We agree that EF1 is useful to understand the risk scoring; however, as each figure is equivalent to a fairly large amount of text and our main text must be shortened, regrettably EF1 will remain in the Extended Figures section (which at least appears on the same webpage as the main text, not separated to Supplementary Materials).

We now clarify in Methods that 10/9 as the notification threshold “was chosen as part of the intervention deployment, not as part of analysis here” (driven by specificity/sensitivity tradeoffs in detecting contacts at 2 metres' distance).

Some specific questions:

- On p 4, two aggregate summary metrics are introduced, namely the total duration of the exposure and the cumulative risk score, which are extensively used later on. It is stated, that "both [are] aggregated over all exposure windows of the contact." Does this really mean, that they are aggregated over all exposure windows of the contact over the entire time period from April 2021 to February 2022? If so, wouldn't it make more sense to aggregate the exposure windows of shorter time periods for the analysis? Something like this is shown in Figure F1b, but the temporal stratification apparently was done only "by the month when the notifications were received", not for the exposure windows analysed. If so, it is less surprising that "this pattern holds irrespective of season or epidemic wave". (I would indeed have expected changing probabilities of infection for pandemic phases with different dominant SARS-CoV-2 variants and an evolving immune protection through vaccination and naturally acquired immunity.)

Apologies for the confusion: the meaning of aggregating over all exposure windows of the contact was meant to indicate all windows reported by the app of the contact *for a given notification event*. We have clarified the mention of aggregation, and added this near the start of Results: “If a given individual was notified multiple times during our study, each notification is analysed here as though it were of a separate contact.” We have added this in Methods where the grouping of events packets is explained: “This procedure for grouping multiple event packets as being from the same contact is specifically for a single notification event of a given contact: if the same individual is notified multiple times during our study, each notification event is treated as being from a separate individual, with a set of event packets associated to each event. This is necessary due to the absence of unique identifiers for each app user.”

Regarding temporal variation, please see our response to substantive point 3 of Referee 2. The short answer is that there is a clearly visible drop in probability of infection due to the vaccination campaign between January-April 2021, i.e. immediately before the start of the period analysed here.

- The same question (aggregation over which time period?) applies to the section "Empirical estimation of individuals' probability of testing positive from summary statistics" on p 16 (Methods). The previous section states: "When more than one individual with given characteristics was notified on a given day, all event packets that day with those characteristics were discarded." This would introduce a bias (exposure summaries would be systematically undercounted), if exposure windows were aggregated over time periods of several or many days; so perhaps only the exposure windows of an individual over a given day are aggregated (?) But this is not stated anywhere, as far as I saw. Also, for instance Figure F1 seems to exclude this interpretation, because it shows exposure durations far beyond 24 hours. (NB: The diagrams in row F1a and F1b are difficult to compare, because their x-axes are differently scaled. Even more so, as these are logarithmic scales, which btw is not mentioned in the label text. It seems that the max duration in F1a exceeds that in F1b, while the cumulative risk score is simply more compressed in F1a than in F1b.)

The same answer applies to the same question (the point immediately above). We have standardised x axes as previously suggested.

- On p 10, household contacts are defined as "exposed for at least 8 hours in a day". Does this mean, that each "contact" who had a cumulated exposure of at least 8 hours on a single day (even without any exposure on any other day) qualifies as a household contact? (This question assumes that exposure windows are aggregated over time periods longer than a day.)

The referee is right. The choice of 8 hours is due to the fact that 7.5 hours is the typical working day in England and Wales. Our categorisation will sometimes be incorrect, but it is "intended to approximately reflect different contexts" given the absence of any contextual data recorded by the app.

The referee is also correct in suspecting that household contacts tend to be exposed across multiple days (even if this is not implied by their definition). In fact, exposures of household contacts are most likely to span 3 days, vs 1 day for non-household contacts. Only 10% of household contacts span a single day, vs 85% of non-household contacts. Furthermore, for those household contacts that span <3 days, the days are most likely to belong to a weekend. These findings are consistent with our expectations if the interpretation given in this paper would be broadly correct.

Specific observations:

- On p 3: "with custom analysis of Bluetooth signal strengths exchanged between nearby devices to estimate their proximity." -> Proximity estimation is done by signal attenuation rather than strength.

Corrected as suggested.

- re. Supplementary Figure SF4 the label states: "the same as Figure SF3, but with observed probability of transmission on a linear rather than a logarithmic scale." However, the opposite seems to apply: SF3 = linear; SF4 = logarithmic.

We removed that figure.

Best wishes on behalf of all co-authors,
Professor Christophe Fraser & Dr Luca Ferretti

Reviewer Reports on the First Revision:

Referees' comments:

Referee #1 (Remarks to the Author):

Thank you for addressing my questions. The paper has gained clarity, and I look forward to seeing it in print. I have no further questions or remarks.

Referee #2 (Remarks to the Author):

The authors have addressed my concerns, and I have no further ones.

Referee #3 (Remarks to the Author):

I thank the authors for extensively responding to the referees' comments and questions and for incorporating most suggestions made into the reworked version of the manuscript. The description of the methodology is very comprehensive and clear now and this impacts greatly on the clarity and value of the results.

I have very few and minor remaining comments.

#1 I am not convinced of the arguments for using the same term "contact" for both, the event and the individual. I acknowledge that this is commonly done and that it might be standard in the UK. (The cited guidance (<https://www.gov.uk/guidance/nhs-test-and-trace-how-it-works>) indeed provides such a definition ("A 'contact' is a person you've been in close contact with if you've tested positive for COVID-19."), but at the same time uses 'contact' continuously in its alternative meaning (as event) throughout that same document ("If ... you've been in close contact with someone ..."; "If you have had close contact with someone ..."; ...).)

While the intended meaning can often be inferred from the context, this is not always the case. In the manuscript, the ambiguity starts with the header: "from 7 million contacts". And - notwithstanding the authors' clarification of their meaning of "contact" in the abstract (see also reference to line 24 below) - one has to read well beyond the abstract to understand what is meant and counted.

A specific reason for emphasizing the distinction is that manual contact tracing traces contact persons, while digital contact tracing (at least with the GAEN framework) traces contact events.

Having said that I can accept the authors' choice, especially with consideration of the word count.

A minor note: In line 24 the authors inserted "(i.e. individuals known to have been exposed to the pathogen)" after "contacts of confirmed cases". However, it may not be entirely clear to every reader,

whether this refers to the confirmed cases themselves or to their contacts.

#2 Given that contacts are defined as individuals exposed to cases, it seems tautological to later on mention "contacts exposed to cases" (see for instance lines 71, 89) or "exposed contacts" (line 278).

#3 In line 81 (first occurrence of the topic) the mention "through the app" (for reporting a positive test) has been deleted. However, this is important for understanding the process. And it has not been deleted at later occurrences (e.g. lines 119/120, 151). This gives room to doubts, whether there may be several ways of reporting a positive test that have been analyzed. Better to fully describe the process in the first place and abbreviate thereafter.

#4 In line 108 the authors explain: "When a contact was notified, their app sent anonymous exposure data to the central server." This gives the impression that the data ('event packets') might be sent immediately upon notification. In line 397 they explain: "Each correctly functioning installation of the app sent one 'analytics packet' of data daily. Each packet indicated whether or not the app user was notified of risky exposure on that day." To me it is still not entirely clear, whether cumulative data (i.e., 1 analytics packet, plus n event packages (1 per exposure window), if applicable) is sent once daily (at the end of the day) to the server or whether notification-triggered event packet submissions can occur at any time. Finally, in line 446 the authors mention "event packets sent on the same day", but they don't explain (unless I missed it) how this "same day" is established (assuming that the exposure windows are usually from previous days). By date (and time?) information included in the event packet or by reception information logged by the server?

I leave it to the discretion of the authors, whether they want to consider these remaining comments. They are not essential for my recommendation to publish the manuscript.

Justus Benzler

Author Rebuttals to First Revision:

Referee #3 (Remarks to the Author)

#1 I am not convinced of the arguments for using the same term "contact" for both, the event and the individual. I acknowledge that this is commonly done and that it might be standard in the UK. (The cited guidance (<https://www.gov.uk/guidance/nhs-test-and-trace-how-it-works>) indeed provides such a definition ("A 'contact' is a person you've been in close contact with if you've tested positive for COVID-19."), but at the same time uses 'contact' continuously in its alternative meaning (as event) throughout that same document ("If... you've been in close contact with someone ..."; "If you have had close contact with someone ..."; ...).) ...

While the intended meaning can often be inferred from the context, this is not always the case... A specific reason for emphasizing the distinction is that manual contact tracing traces contact persons, while digital contact tracing (at least with the GAEN framework) traces contact events.

We apologise for the confusion. In our previous response we said:

- > In the term “contact tracing”, digital or manual, both common meanings of “a contact”—an exposure event or an exposed individual—make sense. For the former, we are talking about a tracing process that is based on such events; for the latter, we are talking about a process that traces such individuals.

Our use of “we” above referred to the community, not to us the authors. This was not intended as a defence of using both meanings interchangeably in the same context, but as a defence of our choosing the other meaning rather than the referee’s preference. In this paper we consistently use “contact” to refer to a person, and describe events as exposure events (except for one instance in main text and a few in supplementary, which we have now corrected).

In the manuscript, the ambiguity starts with the header: "from 7 million contacts". And - notwithstanding the authors' clarification of their meaning of “contact” in the abstract (see also reference to line 24 below) - one has to read well beyond the abstract to understand what is meant and counted...

A minor note: In line 24 the authors inserted "(i.e. individuals known to have been exposed to the pathogen)" after "contacts of confirmed cases". However, it may not be entirely clear to every reader, whether this refers to the confirmed cases themselves or to their contacts.

The character limit of the title does not permit replacing ‘contact’ with an explanation of what this term means. We have now clarified the short definition in the abstract at line 24 to avoid confusion, as suggested. We have now added a precise definition of ‘contact’ (as well as ‘case’ and ‘index case’) in the Introduction.

#2 *Given that contacts are defined as individuals exposed to cases, it seems tautological to later on mention "contacts exposed to cases" (see for instance lines 71, 89) or "exposed contacts" (line 278).*

We agree and have corrected this.

#3 *In line 81 (first occurrence of the topic) the mention "through the app" (for reporting a positive test) has been deleted. However, this is important for understanding the process. And it has not been deleted at later occurrences (e.g. lines 119/120, 151). This gives room to doubts, whether there may be several ways of reporting a positive test that have been analyzed. Better to fully describe the process in the first place and abbreviate thereafter.*

We agree and have corrected this.

#4 *In line 108 the authors explain: "When a contact was notified, their app sent anonymous exposure data to the central server." This gives the impression that the data ('event packets') might be sent immediately upon notification. In line 397 they explain: "Each correctly functioning installation of the app sent one 'analytics packet' of data daily. Each packet indicated whether or not the app user was notified of risky exposure on that day." To me it is still not entirely clear, whether cumulative data (i.e., 1 analytics packet, plus n event packages (1 per exposure window), if applicable) is sent once daily (at the end of the day) to the server or whether notification-triggered event packet submissions can occur at any time.*

We have clarified this.

Finally, in line 446 the authors mention "event packets sent on the same day", but they don't explain (unless I missed it) how this "same day" is established (assuming that the exposure windows are usually from previous days). By date (and time?) information included in the event packet or by reception information logged by the server?

We have clarified this.